# Reinforcement Learning with Logarithmic Regret and Policy Switches

**Grigoris Velegkas**
Yale University
grigoris.velegkas@yale.edu

**Zhuoran Yang**
Yale University
zhuoran.yang@yale.edu

**Amin Karbasi**
Yale University, Google Research
amin.karbasi@yale.edu

## Abstract

In this paper, we study the problem of regret minimization for episodic Reinforcement Learning (RL) both in the model-free and the model-based setting. We focus on learning with general function classes and general model classes, and we derive results that scale with the *eluder dimension* of these classes. In contrast to the existing body of work that mainly establishes instance-independent regret guarantees, we focus on the instance-dependent setting and show that the regret scales logarithmically with the horizon $T$, provided that there is a gap between the best and the second best action in every state. In addition, we show that such a logarithmic regret bound is realizable by algorithms with $O(\log T)$ switching cost (also known as *adaptivity complexity*). In other words, these algorithms rarely switch their policy during the course of their execution. Finally, we complement our results with lower bounds which show that even in the tabular setting, we cannot hope for regret guarantees lower than $O(\log T)$.

## 1 Introduction

The main goal of Reinforcement Learning (RL) is the design and analysis of algorithms for automated decision making in complex and unknown environments. The environment is modeled as a *state space* and the available decisions are modeled as an *action space*. In recent years, RL has seen tremendous success in practical applications including, but not limited to, games and robotics [MKS+15, SHM+16, DCH+16, SSS+17, VBC+19]. Despite this success, a theoretical understanding of the algorithms that are deployed in these settings remains elusive. Traditionally, theoretical RL approaches have focused on the *tabular* setting where the complexity of the algorithms depends on the cardinality of the aforementioned spaces [SB18]. Thus, they are not suitable for applications where the state-action space is very large.

A different approach that has gained a lot of attention recently is the *function approximation* regime where the cumulative reward of the algorithm is modeled through a function, such as linear functions over some feature space. The advantage of this approach is that the algorithm can perform its search over a lower-dimensional space. There is a long line of work that provides regret guarantees for RL in the function approximation setting [OVR14, OVRW16, YW20, JYWJ20, AJS+20, CYJW20, KKL+20, ZBB+20, HZG21, KSWY21, ZHG21, ICN+21].

36th Conference on Neural Information Processing Systems (NeurIPS 2022).

Most of these works have focused on establishing worst-case $\sqrt{T}$-regret guarantees, where $T$ is the number of interactions with the environment. The caveat with these guarantees is that they are pessimistic since they neglect benign settings where even an *exponential* improvement over these bounds is achievable. To address this issue, there are some works that obtain instance-dependent regret bounds for RL in the tabular setting and in the linear function approximation setting [SJ19, YW20, HZG21]. However, getting logarithmic regret bounds in the context of *general* function approximation remains open. Hence, a natural question is the following:

*Can we establish instance-dependent logarithmic regret bounds with general function approximation?*

In this paper, we provide an affirmative answer to this question. Following the assumptions in the existing literature, we model the RL problem as an MDP that enjoys the property that the optimal *policies* are at least $\mathrm{gap}_{\min}$ better than the rest, where $\mathrm{gap}_{\min} > 0$ is a parameter that captures the hardness of the underlying problem. We focus both on the *model-free* and the *model-based* settings with general function approximation, where we represent the value function or the transition model by a given function class, respectively.

In the model-free setting, we study the algorithm proposed in [KSWY21], which is a variant of the least-squares value iteration (LSVI) with upper confidence bound (UCB) bonuses that guide exploration. Here, the bonus functions are given by the width of a data-dependent confidence region for LSVI. For the model-based setting, we develop a similar algorithm which combines value-targeted regression with UCB bonuses. On top of the logarithmic regret guarantees they enjoy, our algorithms feature lazy policy updates, in the sense that the policy is updated rarely and only when certain conditions are met. For both settings, we establish $O(\mathrm{poly}(\log T) \cdot \mathrm{poly}(H) \cdot \mathrm{poly}(d_{\mathcal{F}}) \cdot 1/\mathrm{gap}_{\min})$ regret guarantees, where $T$ is the number of interactions with the environment, $H$ is the *planning horizon*, $d_{\mathcal{F}}$ is a term that captures the complexity of the function class $\mathcal{F}$ which is used to approximate either the value function or the transition model. In particular, $d_{\mathcal{F}}$ involves both the *eluder dimension* [RVR13] and the log-covering numbers of the function classes. That is, for benign MDPs these RL algorithms achieve logarithmic regret, which is exponentially better than the worst-case $O(\sqrt{T})$-regret. Moreover, we extend the result of [KSWY21] regarding the *adaptivity complexity* of the algorithm, i.e. the number of different policies it uses, to the model-based and show that it is logarithmic in $T$. We remark that in many real-world applications, like medical trials and personalized recommendation, there is a high cost and overhead associated with changing the underlying policy. Thus, coming up with algorithms that have low adaptivity complexity is an important task. To the best of our knowledge, this is the first work that establishes a logarithmic instance-dependent regret guarantee for RL with general function approximation.

## 1.1 Related Work

**Logarithmic regret bounds for bandits.** There is a long line of work that establishes logarithmic regret guarantees in bandit problems. Essentially, bandits are a special case of RL where the transition to the next state does not depend on the action that was taken by the agent. An extensive list of such algorithms can be found in [BCB12, Sli19, LS20].

**Logarithmic regret bounds for RL.** A series of works are devoted to proving instance-dependent logarithmic regret bounds in tabular RL. [OPT18, SJ19] prove lower bounds that show that a logarithmic dependence on $T$ is unavoidable. Considering the upper bounds, [AO07, TB07] establish logarithmic regret guarantees in the average reward setting. Both of these guarantees are asymptotic as they require the number of interactions $T$ with the MDP to be large enough. Regarding non-asymptotic bounds, [JOA10] provide such an algorithm that achieves $O(D^2|\mathcal{S}|^2|\mathcal{A}|\log(T)/\mathrm{gap}_{\min})$ regret for the average-reward MDP, where $D$ is the diameter of the MDP. For episodic MDPs, logarithmic regret upper bounds are established in [SJ19, YYD21]. The work that is probably the most closely related to ours is [HZG21]. It provides instance-dependent logarithmic regret guarantees both in the model-free and model-based setting with *linear* function approximation. Moreover, the proposed algorithms update the policy in every episode. Our work generalizes these results since the linear regime is a special case of the setting we are studying. Furthermore, we achieve an exponential improvement on the adaptivity complexity over their algorithms.

**Bandits with limited adaptivity complexity.** There is a lot of interest in obtaining bandit algorithms that update their policies rarely [AYPS11, PRCS16, AAAK17, GHRZ19, DLZZ20, RYZ21]. Notably, [DLZZ20] study rare policy switching constraints for a broader class of online learning and decision making problems such as logit bandits.

**RL with limited adaptivity complexity.** Recently, there has been a lot of interest in developing RL algorithms that achieve sub-linear regret and have low adaptivity complexity [BXJW19, ZZJ20, CK20, WZG21, KSWY21, GXDY21]. We develop an algorithm with low adaptivity complexity that works in the model-based, general function approximation setting.

**RL with general function approximation.** As we have alluded to already, it is important to develop and analyze algorithms in the function approximation regime. So far, the most commonly studied setting is RL with linear function approximation [YW20, JYWJ20, DLMW20, WSY20, ZLKB20, AKKS20]. Recently, there are also important results in RL with general function approximation. To be specific, [JKA+17] design an efficient algorithm whose sample complexity is bounded in terms of the Bellman rank of the function class. [AJS+20] develop an algorithm for model-based RL, whose regret bound depends on the eluder dimension of the underlying class of models. [JLM21] propose an algorithm for problems where the underlying class has bounded Bellman eluder dimension. Recent works such as [WSY20, KSWY21, FRSLX20] develop algorithms in the model-free setting whose regret scale with the eluder dimension of the functions.

## 2 Preliminaries

### 2.1 Notation

We use the notation $[N] = \{1, 2, \ldots, N\}$. We also define the infinity norm of some function $f : \mathcal{X} \to \mathbb{R}$, where $\mathcal{X}$ is some domain, to be $||f||_\infty = \sup_{x \in \mathcal{X}} |f(x)|$. For a dataset $\mathcal{D} = \{(x_i, q_i)\}_{i=1}^n \subseteq \mathcal{X} \times \mathbb{R}$ and a function $f : \mathcal{X} \to \mathbb{R}$, we define the following norm $||f||_\mathcal{D} = \left(\sum_{i=1}^n (f(x_i) - q_i)^2\right)^{1/2}$. Given a set $\mathcal{Z} = \{x_i\}_{i=1}^n \subseteq \mathcal{X}$ we let $||f||_\mathcal{Z} = \left(\sum_{i=1}^n f(x_i)^2\right)^{1/2}$ be the data-dependent norm. Given a measurable set $\mathcal{X}$, we denote with $\Delta(\mathcal{X})$ the probability simplex over $\mathcal{X}$. We also denote by $\mathbb{1}[\mathcal{E}]$ the indicator function of the event $\mathcal{E}$ and by $\mathrm{poly}(x)$ a function that is a polynomial in $x$.

### 2.2 Episodic Markov Decision Processes

The learning agent interacts with the environment over a sequence of $K$ rounds which we call *episodes*. We model the interaction of the agent with the environment in every episode as a *Markov Decision Process* (MDP). We denote an MDP by $M = (\mathcal{S}, \mathcal{A}, P, r, H, s_1)$, where $\mathcal{S}$ is the *state space*, $\mathcal{A}$ is the *action space*, $P = \{P_h : \mathcal{S} \times \mathcal{A} \to \Delta(\mathcal{S})\}_{h=1}^H$ are the *transition kernels*, $r = \{r_h : \mathcal{S} \times \mathcal{A} \to [0, 1]\}_{h=1}^H$ are the *reward functions* which we assume to be deterministic, $H$ is the *planning horizon*, which is the length of every episode, and $s_1$ is the initial state of every episode. We let $T = K \cdot H$ be the total number of interactions with the MDP. During every episode, the agent uses a *policy* $\pi = \{\pi_h : \mathcal{S} \to \mathcal{A}\}_{h=1}^H$, to take an action at a given state. We use the Q-function and V-function to evaluate the expected total reward generated by a policy $\pi$. More specifically, we define

$$Q_h^\pi(s, a) = \mathbb{E}\left[\sum_{h'=h}^H r_{h'}(s_{h'}, a_{h'}) \big| s_h = s, a_h = a, \pi\right], \quad V_h^\pi(s) = \mathbb{E}\left[\sum_{h'=h}^H r_{h'}(s_{h'}, a_{h'}) \big| s_h = s, \pi\right],$$

where the actions are picked according to $\pi$ and $s_{h'+1} \sim P_{h'}(\cdot | s_{h'}, a_{h'})$. For simplicity, we denote $\langle P_h(\cdot | s, a), V \rangle = \mathbb{E}_{s' \sim P_h(\cdot | s, a)}[V(s')]$. We denote the optimal policy for a given MDP with $\pi^*$. For the optimal Q-function, V-function we use $Q_h^*(s, a) = Q_h^{\pi^*}(s, a), V_h^*(s) = V_h^{\pi^*}(s)$, respectively.

The goal of the learner is to improve its performance as it interacts with the unknown environment. In the episodic setting, the agent commits to a policy at the beginning of every episode. We let $\pi^k$ denote the policy that the agent uses in the $k$-th episode. At each step $h \in [H]$, the agent observes the state $s_h^k$, chooses an action according to $\pi^k$, and then observes the reward $r_h(s_h^k, a_h^k)$ and the next state $s_{h+1}^k \sim P_h(\cdot | s_h^k, a_h^k)$. In this work, to measure the performance of the agent we use the notion of *regret*, defined as

$$\mathrm{Regret}(K) = \sum_{k=1}^K \left(V_1^*(s_1) - V_1^{\pi^k}(s_1)\right).$$

The regret measures the difference between the total reward that the agent would have accumulated if she was following the optimal policy and the reward she actually accumulates. We strive for

algorithms that guarantee sub-linear regret, because, as $K \to \infty$, the reward of the algorithm approaches that of the the optimal policy.

An assumption we make in order to achieve logarithmic regret guarantees is that the minimum *sub-optimality* gap is positive.

**Definition 2.1.** We define the sub-optimality gap of a state-action pair $(s, a)$ at step $h$ to be

$$\text{gap}_h(s, a) = V_h^*(s) - Q_h^*(s, a).$$

The minimum sub-optimality gap is defined to be

$$\text{gap}_{\min} = \inf_{h,s,a} \{\text{gap}_h(s, a) : \text{gap}_h(s, a) \neq 0\}.$$

It is well-known that if we do not make any assumptions the best regret guarantee we can hope for is $O(\sqrt{T})$ [JOA10]. In this work, we derive instance-dependent regret guarantees that achieve an exponential improvement on $T$ when $\text{gap}_{\min} > 0$. We remark that this assumption is *not* restrictive and it rules out some pathological cases. In particular, it allows us to handle MDPs with multiple optimal policies. For a detailed discussion, we kindly refer the reader to Appendix A.1.

## 2.3 Model-Free Assumption

In this paper, we deal with general function classes. In the model-free setting we assume that we have access to a function class $\mathcal{F} \subseteq \{f : \mathcal{S} \times \mathcal{A} \to [0, H + 1]\}$. Our goal is to use the functions in $\mathcal{F}$ to approximate the optimal Q-function. In order to derive meaningful results we assume that this class has some structure. We follow the same assumption as in [WSY20, KSWY21].

**Assumption 2.2** (Bellman Operator Assumption). For any $h \in [H]$ and $V : \mathcal{S} \to [0, H]$ there exists some $f_V \in \mathcal{F}$ such that for all $(s, a) \in \mathcal{S} \times \mathcal{A}$ we have

$$f_V(s, a) = r_h(s, a) + \sum_{s' \in \mathcal{S}} P_h(s'|s, a)V(s').$$

The intuition behind this assumption is that if we apply the one-step Bellman backup operator to some value function $V$, i.e. $r_h(s, a) + \sum_{s' \in \mathcal{S}} P_h(s'|s, a)V(s')$, the result will remain in the function class. Thus, it implicitly poses some constraints both on the transition probabilities and the reward function. It is known that both the tabular setting and the linear MDP setting satisfy this assumption [YW19, JYWJ20]. Another assumption we make is that the function class and the state-action space have bounded covering numbers. We will show that the dependence of the regret on the covering number is poly-logarithmic. This assumption has also appeared in other works [RVR13, WSY20, JLM21, KSWY21].

**Assumption 2.3** (Bounded Covering Number). We say that $\mathcal{N}(\mathcal{F}, \varepsilon)$ is a bound on the $\varepsilon$-covering number of $\mathcal{F}$, if for any $\varepsilon > 0$ there is an $\varepsilon$-cover $\mathcal{C}(\mathcal{F}, \varepsilon) \subseteq \mathcal{F}$ with size $|\mathcal{C}(\mathcal{F}, \varepsilon)| \leq \mathcal{N}(\mathcal{F}, \varepsilon)$, so that for all $f \in \mathcal{F}$ there is some $f' \in \mathcal{C}(\mathcal{F}, \varepsilon)$ such that $||f - f'||_\infty \leq \varepsilon$. Similarly, we say that $\mathcal{N}(\mathcal{S} \times \mathcal{A}, \varepsilon)$ is a bound on the $\varepsilon$-covering number of $\mathcal{S} \times \mathcal{A}$ with respect to $\mathcal{F}$, if for any $\varepsilon > 0$ there is an $\varepsilon$-cover $\mathcal{C}(\mathcal{S} \times \mathcal{A}, \varepsilon) \subseteq \mathcal{S} \times \mathcal{A}$ with size $|\mathcal{C}(\mathcal{S} \times \mathcal{A}, \varepsilon)| \leq \mathcal{N}(\mathcal{S} \times \mathcal{A}, \varepsilon)$, so that for all $(s, a) \in \mathcal{S} \times \mathcal{A}$ there is some $(s', a') \in \mathcal{C}(\mathcal{S} \times \mathcal{A}, \varepsilon)$ such that $\sup_{f \in \mathcal{F}} |f(s, a) - f(s', a')| \leq \varepsilon$.

The intuition behind this assumption is straightforward: even if the function class or the state-action space are infinite, we can approximate them using a small number of points.

## 2.4 Model-Based Assumption

The assumptions in Section 2.3 are *model-free* since they impose some structure on the function class that approximates the Q-function instead of the transition kernel. In order to derive our results, we can also follow a different route and impose some structure directly on the transition kernel [AJS+20].

**Assumption 2.4** (Known Transition Model Family). For all $h \in [H]$, the transition model $P_h$ belongs to a family of models $\mathcal{P}_h$ which is known to the learner. The elements of $\mathcal{P}_h$ are transition kernels that map state-action pairs to signed distributions over the state space $\mathcal{S}$.

We allow signed distributions in our model class to increase its generality. For example, this is useful when we are given access to a model class that can be compactly represented only when it includes non-probability kernels. For an extensive discussion about this, the reader is referred to [PS16].

Transition kernels have been used to model complex stochastic controlled systems. For instance, transitions in robotics systems are often modelled using parameters of the environment, such as friction. An important class that satisfies this assumption are the *linear mixture models*.

**Definition 2.5.** The class of models $\mathcal{P}$ with feature mapping $\phi(s'|s,a) : \mathcal{S} \times \mathcal{S} \times \mathcal{A} \to \mathbb{R}^d$ and some $\theta^* \in \mathbb{R}^d$ whose euclidean norm is bounded, is called linear mixture model if:

- $P(s'|s,a) = \langle \phi(\cdot|s,a), \theta^* \rangle$.

- For any bounded function $V : \mathcal{S} \to [0,H]$ and any pair $(s,a) \in \mathcal{S} \times \mathcal{A}$, we have $||\phi_V(s,a)||_2 \leq \sqrt{H}$, where $\phi_V(s,a) = \langle \phi(\cdot|s,a), V \rangle$.

One way to interpret the linear mixture model is as an aggregation of some basis models which are known to the designer [MJTS20]. Another interesting way to think about it comes from large-scale queuing networks where both the arrival rate of jobs and the processing speed for the queues are unknown. If we approximate this system in discrete time, then the transition matrix from timestep $t$ to timestep $t + \Delta t$ approaches that of a linear function with respect to the arrival rate and the processing time [GK89]. Another interesting setting that satisfies this assumption is the linear-factored MDP [YW20].

## 2.5 Complexity Measure: Eluder Dimension

Our results depend on the complexity of the function classes and the model classes that we consider. To measure this complexity, we use the *eluder dimension* of these classes [RVR13].

**Definition 2.6.** Fix some $\varepsilon \geq 0$ and a sequence of $n$ points $\mathcal{Z} = \{(x_i)\}_{i \in [n]} \subseteq \mathcal{X}$. Then:

1. A point $x \in \mathcal{X}$ is $\varepsilon$-dependent on $\mathcal{Z}$ with respect to $\mathcal{F}$ if for all $f, f' \in \mathcal{F}$ such that $||f - f'||_{\mathcal{Z}} \leq \varepsilon$ it holds that $|f(x) - f'(x)| \leq \varepsilon$.

2. A point $x$ is $\varepsilon$-independent of $\mathcal{Z}$ with respect to $\mathcal{F}$ if $x$ is not $\varepsilon$-dependent on $\mathcal{Z}$.

3. The $\varepsilon$-eluder dimension of $\mathcal{F}$, which is denoted by $\dim_E(\mathcal{F}, \varepsilon)$, is the length of the longest sequence of elements in $\mathcal{X}$ such that every element in this sequence is $\varepsilon'$-independent of its predecessors, for some $\varepsilon' \geq \varepsilon$.

Intuitively, the eluder dimension of $\mathcal{F}$ quantifies the smallest set of elements $\mathcal{Z} \subseteq \mathcal{X}$ so that if all $f \in \mathcal{F}$ are close with respect to $\mathcal{Z}$, then they are close on all elements of $\mathcal{X}$.

It is known that when $\mathcal{X} = \mathcal{S} \times \mathcal{A}$, $f : \mathcal{S} \times \mathcal{A} \to [0,H]$, and $\mathcal{S}, \mathcal{A}$ are finite, we have that $\dim_E(\mathcal{F}, \varepsilon) \leq |\mathcal{S}| \cdot |\mathcal{A}|$, for all $\varepsilon > 0$ [RVR13, WSY20]. Moreover, when $\mathcal{F}$ is the class of linear functions, i.e., $f_\theta(s,a) = \theta^T \phi(s,a)$, for a given feauture vector $\phi(s,a)$, the eluder dimension of $\mathcal{F}$ is bounded by $\dim_E(\mathcal{F}, \varepsilon) = O(d \log(1/\varepsilon))$, for all $\varepsilon > 0$. We remark that classes which go beyond linear functions have bounded eluder dimension. Some examples are quadratic functions, generalized linear functions (e.g. sigmoids) which have the form $g(\langle \phi(s), x \rangle)$, where $g$ is increasing and Lipshitz-continuous with non-zero derivative, all finite function classes, sparse linear combinations of features, linear thresholds with Gaussian i.i.d. features, etc. [RVR13, OVR14, LKFS21]. It is an ongoing line of work to understand what is the most general class of functions that has bounded eluder dimension. We refer to the setting we are working on as *general function approximation* to align with the previous works that study function classes with bounded eluder dimension.

## 2.6 Switching Cost

Essentially, the *switching cost* or the *adaptivity complexity* measures the number of episodes the algorithm has to update its policy in order to achieve the guaranteed regret bound [BXJW19, KSWY21].

**Definition 2.7.** We define the switching cost of an algorithm $A$ over $K$ episodes to be

$$N_{\text{switch}} = \sum_{k=1}^{K-1} \mathbb{1}[\pi_k \neq \pi_{k+1}].$$

# 3 Overview of the Algorithms and Main Results

In this section, we present our main results and give a high-level description of the techniques we use. We treat both the model-free and the model-based setting in a unified way. The algorithm for the model-free setting comes directly from [KSWY21]. We extend it appropriately in order to handle the model-based setting. The main algorithm is presented in Algorithm 1. The only differences between the two settings are the different sampling routine and Q-function estimator used by Algorithm 1. In a nutshell, the low-switching cost algorithm works as follows:

- After each round of the interaction with the MDP, Algorithm 1 adds elements to the current dataset with some probability that depends on their significance and updates the policy only if the dataset has changed. This guarantees that the adaptivity of the algorithm depends logarithmically on $T$, without hurting the regret guarantee. The sampling routines are postponed to Appendix A.2.

- There is a least-squares estimate of the Q-function (transition kernel) in the model-free (model-based) setting.

- A bonus is added to this estimate which encourages exploration and, with high probability, guarantees that the current estimate of the Q-function serves as an element-wise upper bound of $Q^*$. This bonus is based on a sub-sampled dataset that has been accumulated from previous interactions with the MDP.

Before we delve deeper into the two settings separately, we describe a parameter that is crucial for both of the algorithms we are using. Following [KSWY21], the *sensitivity* of an element $z$ with respect to a dataset $\mathcal{Z}$ and a function class $\mathcal{F}$ is defined to be

$$\text{sensitivity}_{\mathcal{Z},\mathcal{F}}(z) = \min\left\{\sup_{f_1,f_2\in\mathcal{F}}\frac{(f_1(z) - f_2(z))^2}{\min\{\|f_1 - f_2\|_{\mathcal{Z}}^2, T(H+1)^2\} + \beta}, 1\right\}.$$

Intuitively, this parameter captures the importance of the current element $z$ relative to the dataset we are working with. We will elaborate on the choice of the parameter $\beta$ for each of the two settings separately. To establish the regret guarantee, we propose a novel regret decomposition for this algorithm where we utilize the "peeling technique" that has been applied in prior works in local Rademacher complexities [BBM05] and in RL [HZG21, YYD21]. By doing that, we show how the regret of the algorithm relates to the suboptimality gap.

To establish the lower bound on the regret of any algorithm in the settings we are interested in, we utilize a result that was proved in [OPT18, SJ19]. It states that for all algorithms that achieve sublinear regret, there exists a tabular MDP where its regret is at least $\Omega(\text{poly}(\log(T)) \cdot \text{poly}(H) \cdot 1/\text{gap}_{\min})$. We remark that tabular MDPs satisfy both the model-free assumption [WSY20] and the model-based assumption [AJS$^+$20], thus the lower bounds follow immediately from this result.

## 3.1 Model-Free Setting

We first present the approach we use in the model-free setting, i.e., where we have access to some function class $\mathcal{F}$ and state-action space $\mathcal{S} \times \mathcal{A}$ that satisfy Assumption 2.2 and Assumption 2.3, respectively. The algorithm we use comes from [KSWY21]. The Q-function estimator and the sampling routine that we use for this setting are presented in Algorithm 2 and Algorithm 4, respectively. The dataset includes pairs of the form $z_h^k = (s_h^k, a_h^k)$. The Q-function routine is a least-squares estimator that is based on all the previous interactions with the MDP. Notice that the bonus function is based only on the sub-sampled dataset and depends on a hardcoded parameter $\beta$. This parameter is chosen in a way that ensures the Q-function is an optimistic estimate of the actual one and the bonus we add is not too large. The choice of $\beta$ is given in [KSWY21].

A crucial part of the algorithm is the online sub-sampling routine. The reason we are using this routine is twofold. Firstly, if we use the entire dataset there will be a huge number of distinct elements in it, which can make the exploration bonus unstable since it changes constantly and can take infinitely many different values. In order to establish the optimism of the Q-function estimation, namely,

$$Q_h^*(s,a) \le Q_h^k(s,a) \le \langle P_h(\cdot|s,a), V_h^k\rangle + 2b_h^k(s,a),$$

[KSWY21] show that it is crucial to bound the complexity of the exploration bonus. Secondly, if we sub-sample the dataset based on the importance of the elements, we can achieve the regret guarantees that we are aiming for by switching the policy only when an important element has been added. Notice that whenever an element is added to the dataset, multiple copies are included. This is to make the sub-sampled dataset behave like an unbiased estimator of the orignal one. Then, using concentration bounds one can show that it approximates the original one with high probability [KSWY21]. The full description of this procedure is presented in Algorithm 4. For a more detailed discussion about the importance of sub-sampling the dataset, the interested reader is referred to [WSY20, KSWY21].

---

**Algorithm 1** Low Switching Cost Value Iteration (with parameters $\delta, K$)

---

**Require:** Failure probability $\delta \in (0,1)$, number of episodes $K$, and setting of operation
1: $\widetilde{k} \leftarrow 1$
2: $\widehat{Z}_h^1 \leftarrow \emptyset, \forall h \in [H]$
3: **for** $k \in [K]$ **do**
4:     **for** $h = H, H-1, \ldots, 1$ **do**
5:         **if** $k \geq 2$ **then**
6:             $\widehat{\mathcal{Z}}_h^k \leftarrow$ **Sample**$(\mathcal{F}, \widehat{\mathcal{Z}}_h^{k-1}, z_h^{k-1}, \delta)$
7:         **end if**
8:     **end for**
9:     **if** $k = 1$ or $\exists h \in [H]: \widehat{Z}_h^k \neq \widehat{Z}_h^{\widetilde{k}}$ **then**
10:         $\widetilde{k} \leftarrow k$
11:         $Q_{H+1}^k(\cdot,\cdot) \leftarrow 0, V_{H+1}^k(\cdot) \leftarrow 0$
12:         **for** $h = H, H-1, \ldots, 1$ **do**
13:             $\mathcal{T}_h^k \leftarrow$ history of execution
14:             $Q_h^k(\cdot,\cdot) \leftarrow$ **Q-Estimator**$(\mathcal{T}_h^k, \mathcal{Z}_h^k)$
15:             $V_h^k(\cdot) = \max_{a \in \mathcal{A}} Q_h^k(\cdot, a)$
16:             $\pi_h^k(\cdot) \leftarrow \arg\max_{a \in \mathcal{A}} Q_h^k(\cdot, a)$
17:         **end for**
18:     **end if**
19:     Receive initial state $s_1$ of episode $k$
20:     **for** $h \in [H]$ **do**
21:         Take action $a_h^k \leftarrow \pi_h^{\widetilde{k}}(s_h^k)$
22:     **end for**
23: **end for**

---

**Algorithm 2** Q-function Model-Free Estimator

---

**Require:** Current sub-sampled dataset $\widehat{\mathcal{Z}}$, history of execution $\mathcal{T}$
1: $\mathcal{D}_h^k \leftarrow \{(s_h^\tau, a_h^\tau, r_h^\tau + V_{h+1}^k(s_{h+1}^\tau))\}_{\tau \in [k-1]}$
2: $\widehat{f} \leftarrow \arg\min_{f \in \mathcal{F}} ||f||_{\mathcal{D}}^2$
3: $\widehat{F}_h^k \leftarrow \{f_1, f_2 \in \mathcal{F} : \min\{||f_1 - f_2||_{\widehat{\mathcal{Z}}_h^k}^2, T(H+1)^2 \leq \beta\}$
4: $b_h^k(\cdot,\cdot) \leftarrow \sup_{f_1,f_2 \in \widehat{F}_h^k} |f_1(\cdot,\cdot) - f_2(\cdot,\cdot)|$
5: **Return** $\min\{f_h^k(\cdot,\cdot) + b_h^k(\cdot,\cdot), H\}$

---

**Algorithm 3** Q-function Model-Based Estimator

---

**Require:** Function class $\mathcal{F}$, current sub-sampled dataset $\widehat{\mathcal{Z}}$, current regression dataset $\mathcal{D}$
1: $\widehat{P}_h^k \leftarrow \arg\min_{P \in \mathcal{P}_h}$
$\sum_{k'=1}^{k-1} \left( \langle P(\cdot|s_h^{k'}, a_h^{k'}), V_{h+1}^{k'} \rangle - V_{h+1}^{k'}(s_{h+1}^{k'}) \right)^2$
2: $\mathcal{F}_h^k = \{f_1, f_2 : \min\{||f_1 - f_2||_{\widehat{\mathcal{Z}}_h^k}, T(H+1)^2\} \leq \beta\}$
3: $b_h^k(\cdot,\cdot) \leftarrow \sup_{f_1,f_2 \in \mathcal{F}_h^k} |f_1(\cdot,\cdot,V_{h+1}) - f_2(\cdot,\cdot,V_{h+1})|$
4: **Return** $\min\{r_h(\cdot,\cdot) + \langle \widehat{P}_h^k(\cdot|\cdot,\cdot), V_{h+1}^k \rangle + b_h^k(\cdot,\cdot), H\}$

---

We are now ready to state our main result in this setting.

**Theorem 3.1.** *There exists an absolute constant $C > 0$, and a proper parameter $\beta$ for Algorithm 1 such that with probability of at least $1 - \lceil \log T \rceil e^{-\tau} - \delta$ the regret of the algorithm is bounded by*

$$\mathrm{Regret}(K) \leq \frac{Cd_{\mathcal{F}} H^5 \log^4 T}{\mathrm{gap}_{\min}} + \frac{16 H^2 \tau}{3} + 2,$$

*for any $\delta, \tau > 0$, where $d_{\mathcal{F}} = \dim_E^2(\mathcal{F}, 1/T) \cdot \log(\mathcal{N}(\mathcal{F}, \delta/T^2)/\delta) \cdot \log(\mathcal{N}(\mathcal{S} \times \mathcal{A}, \delta/T^2)/\delta)$ is a parameter that captures the complexity of the function class. The value of the parameter $\beta$ is*

$$\beta = Cd_{\mathcal{F}}' H^2 \log^4 T,$$

*where $d_{\mathcal{F}}' = \log(\mathcal{N}(\mathcal{F}, \delta/T^3)/\delta) \dim_E(\mathcal{F}, 1/T) \log(\mathcal{N}(\mathcal{S} \times \mathcal{A}, \delta/T^3)/\delta))$. Moreover, the number of switching policies is bounded by*

$$O\left(H \log(T\mathcal{N}(\mathcal{F}, \sqrt{\delta}/T^2)/\delta) \dim_E(\mathcal{F}, 1/T) \log^2 T\right).$$

## 3.2 Model-Based Setting

In this regime, we assume that the MDP satisfies Assumption 2.4. We also assume that the reward function is known to the learner similar to [AJS+20]. If the reward is unknown, we just estimate it and construct a confidence region.

Before we discuss the details of our approach, we need to describe an important set of functions that show up in our algorithm and in the regret guarantee. Let $\mathcal{V}$ be the set of all measurable functions that are bounded by $H$. Let $\mathcal{P}_h$ be the set of potential models in step $H$. We also let $f : \mathcal{S} \times \mathcal{A} \times \mathcal{V} \to \mathbb{R}$ and define the following set:

$$\mathcal{F}_h = \left\{ f : \exists \widetilde{P}_h \in \mathcal{P}_h \text{ so that } f(s, a, V) = \int_{\mathcal{S}} \widetilde{P}_h(s'|s, a) V(s') ds', \forall (s, a, V) \in \mathcal{S} \times \mathcal{A} \times \mathcal{V} \right\}.$$

The bounds we state scale with the complexity of $\mathcal{F}_h$. The Q-function estimator and the sampling routine we use that are specific to this setting are presented in Algorithm 3 and Algorithm 5, respectively. The dataset includes elements of the form $z_h^k = (s_h^k, a_h^k, V_{h+1}^k(\cdot))$. The Q-function routine works in the following way. We first estimate a model $\widehat{P}_h \in \mathcal{P}_h$ using a least-squares estimator that is based on all the previous interactions with the MDP. Using a concentration argument for this estimator of the model, similar to [RVR13, AJS$^+$20], we can show that for an appropriate choice of $\beta$, the estimated model lies in a data-dependent ball centered at $\widehat{P}_h$, with high probability (see Lemma C.11 in the Appendix). Thus, we can set the bonus function to be the diameter of this ball in order to ensure that $\widehat{Q}_h$ is an optimistic estimate of $Q_h^*$. In addition, the choice of $\beta$ ensures that the bonus we add is not very large. Notice also that since the concentration argument in this setting differs with that in the model-free setting, we do not need to round the elements that we are adding to the sub-sampled dataset. In this setting, the main reason we sub-sample the dataset is to achieve logarithmic adaptivity. To bound the adaptivity complexity, we use a similar approach as in [KSWY21].

We are now ready to state our main result in this setting.

**Theorem 3.2.** *There exists an absolute constant $C$ and a proper value of the parameter $\beta$ for Algorithm 1 such that with probability at least $1 - \lceil \log T \rceil e^{-\tau} - \delta$ the regret of the algorithm is bounded by*

$$\text{Regret}(K) \leq \frac{C d_{\mathcal{F}} H^5 \log T}{\text{gap}_{\min}} + \frac{16 H^2 \tau}{3} + 2,$$

*where $d_{\mathcal{F}} = \log(\mathcal{N}(\mathcal{F}, 1/T)/\delta) \dim_E^2(\mathcal{F}, 1/T)$. The value of the parameter $\beta$ is*

$$\beta = 4H^2 \log(2\mathcal{N}(\mathcal{F}, 1/T)/\delta) + 4/H \left( C + \sqrt{H^2/4 \log(T/\delta)} \right),$$

*where $\mathcal{N}(\mathcal{F}, 1/T) = \max_{h \in [H]} \mathcal{N}(\mathcal{F}_h, 1/T), \dim_E(\mathcal{F}, 1/T) = \max_{h \in [H]} \dim_E(\mathcal{F}_h, 1/T)$.*

*The number of switching policies is bounded by*

$$O\left( H \log(T \mathcal{N}(F, \sqrt{\delta/(64T^3)})/\delta) \dim_E(\mathcal{F}, 1/T) \log^2 T \right).$$

We remark that the eluder dimension and the log-covering numbers, which show up in both of our bounds, depend on $T$. To the best of our knowledge, in all the known cases that the eluder dimension is bounded (e.g. linear functions, quadratic functions, generalized linear functions), the scaling is $\text{poly}(\log T)$ (see e.g. [RVR13]).

### 3.3 Technical Contributions

We briefly highlight the main technical contributions of this work. Regarding algorithmic contributions, we extend the algorithm from [KSWY21], which works in the model-free setting, to the model-based setting. Even though these two algorithms look similar, the approach to prove concentration, optimism, and correctness in the model-based setting is different (see, e.g., Lemma C.12). This perspective provides a unified treatment of both settings that can be beneficial and leads to new results. For instance, if we wish to get instance-independent regret guarantees using the standard regret decomposition, we can get $\sqrt{T}$-regret with $\text{poly}(\log T)$ adaptivity complexity in the model-based setting. This is an *exponential* improvement in the adaptivity complexity compared to [AJS$^+$20]. Furthermore, the analysis to get the instance-dependent guarantees of both algorithms diverges significantly from [KSWY21] since we use a different regret decomposition. Logarithmic regret guarantees in the linear function approximation setting have been established in [HZG21]. However, the technical results that are needed to derive the guarantees in this setting are tailor-made to linear

functions. Importantly, [HZG21] bound several quantities in terms of $\phi(s,a)^\top (\Lambda_h^k)^{-1}\phi(s,a)$ where $\phi(s,a)$ are the features of the state-action pair $(s,a)$ and $\Lambda_h^k = \sum_{i=1}^{k-1}\phi(s_h^i,a_h^i)\phi(s_h^i,a_h^i)^\top + \lambda I$. Since we are working with general functions, we need to derive bounds that scale with the *eluder dimension* of these classes (see, e.g. Lemma C.13).

**Remark 3.3** (Computational Complexity). As it argued in [KSWY21], the main computational bottleneck of the algorithms is solving a (weighted) least squares problem over some set. Given such a procedure, one can also construct the confidence region and estimate the exploration bonus and the sensitivity. [KSWY21] shows that the computational complexity of the algorithm overall is $O\left(\text{poly}(d_{\mathcal{F}} H \log K)\right)$. Importantly, if there is some structure on the space and solving the least-squares problem becomes easier then the computational complexity of the algorithm drops. We underline that solving a least-squares problem is an important component of most theoretical RL algorithms that we are aware of which go beyond the tabular setting. A similar result holds in the model-based setting as well.

## 4 Proof Sketch of Main Results

In this section, we sketch the proof of our main results. Due to space limitation, we only discuss the model-free setting. The full proofs in the model-free, model-based setting can be found in Appendix B, Appendix C, respectively.

The first step in our analysis is the regret decomposition of the algorithm. Lemma B.1 shows that $\mathbb{E}\left[\text{Regret}(K)\right] = \mathbb{E}\left[\sum_{k=1}^{K}\sum_{h=1}^{H} \text{gap}_h(s_h^k, a_h^k)\right]$. Thus, we see that to bound the regret it is enough to bound $\sum_{k=1}^{K} \text{gap}_h(s_h^k, a_h^k)$ for every $h \in [H]$. Towards this end, notice that $\text{gap}_h(s_h^k, a_h^k) = 0$ or $\text{gap}_h(s_h^k, a_h^k) \in [\text{gap}_{\min}, H]$. We apply the "peeling technique" that has also been used in local Rademacher complexities [BBM05] and in [HZG21, YYD21]. The idea is to split the interval $[0, H]$ into $\log(H/\text{gap}_{\min})$ intervals, where the $i$-th interval is $[2^{i-1}\text{gap}_{\min}, 2^i\text{gap}_{\min}]$. Hence, for every $\text{gap}_h(s_h^k, a_h^k)$ that falls in the $i$-th interval its contribution to the regret is at most $2^i\text{gap}_{\min}$. Thus, to bound the regret it suffices to bound the number of suboptimalities that fall into every interval. Notice that for some $\text{gap}_h(s_h^k, a_h^k)$ in this interval we have that

$$V_h^*(s_h^k) - Q_h^{\pi_k}(s_h^k, a_h^k) \geq \text{gap}_h(s_h^k, a_h^k) \geq 2^{i-1}\text{gap}_{\min},$$

so it suffices to bound the number of sub-optimalities $V_h^*(s_h^k) - Q_h^{\pi_k}(s_h^k, a_h^k)$ that fall into the $i$-th interval. Both for the model-free and the model-based setting, we can derive such a bound. Finally, notice that once we have bounded the number of suboptimalities in every interval, it is not difficult to bound the total regret. Let $C_i = [2^{i-1}\text{gap}_{\min}, 2^i\text{gap}_{\min})$ and $N = \log(H/\text{gap}_{\min})$. Then, we know that $\text{Regret}(K)$ can be upper bounded by

$$\sum_{k=1}^{K}\sum_{h=1}^{H} \text{gap}_h(s_h^k, a_h^k) = \sum_{i=1}^{N}\sum_{\text{gap}_h(s_h^k, a_h^k) \in C_i} \text{gap}_h(s_h^k, a_h^k) \leq \sum_{i=1}^{N}\sum_{k=1}^{K} 2^i \mathbb{1}\left[\text{gap}_h(s_h^k, a_h^k) \in C_i\right]$$

$$\leq \sum_{i=1}^{N} 2^i \sum_{k=1}^{K} \mathbb{1}\left[V_h^*(s_h^k) - Q_h^{\pi_k}(s_h^k, a_h^k) \geq 2^{i-1}\text{gap}_{\min}\right].$$

Hence, deriving the regret guarantee boils down to bounding

$$\sum_{k=1}^{K} \mathbb{1}\left[V_h^*(s_h^k) - Q_h^{\pi_k}(s_h^k, a_h^k) \geq 2^i\text{gap}_{\min}\right].$$

We provide such a bound in Lemma B.9, which depends polynomially on $\log T, 1/\text{gap}_{\min}$, and the complexity parameters of the function class. The outline of the proof is the following. We fix some episode $h \in [H]$ and let $K'$ be the set of rounds where $V_h^*(s_h^k) - Q_h^{\pi_k}(s_h^k, a_h^k) \geq 2^i\text{gap}_{\min}$. To get a bound on $|K'|$, our approach is to lower bound and upper bound the quantity $\sum_{i=1}^{|K'|} Q_h^{k_i}(s_h^{k_i}, a_h^{k_i}) - Q_h^{\pi_{k_i}}(s_h^{\pi_{k_i}}, a_h^{\pi_{k_i}})$ by $f_1(|K'|), f_2(|K'|)$, respectively. Then, we use the fact that $f_1(|K'|) \leq f_2(|K'|)$ to establish our bound. For the lower bound, using the definition of $K'$ we get $f_1(|K'|) = 2^i\text{gap}_{\min}|K'|$. For the upper bound, we leverage the fact that $Q_h^{k_i}(s_h^{k_i}, a_h^{k_i}) \leq \langle P_h(\cdot|s_h^{k_i}, a_h^{k_i}), V_{h+1}^{k_i}\rangle + 2b_h^{k_i}(s_h^{k_i}, a_h^{k_i})$ (cf. Lemma B.8) and obtain $f_2(|K'|) \leq$

$\sum_{i=1}^{|K'|} \sum_{h'=h}^{H} \varepsilon_{h'}^{k_i} + \sum_{i=1}^{|K'|} \sum_{h'=h}^{H} b_{h'}^{k_i}(s_{h'}^{k_i}, a_{h'}^{k_i})$, where $\varepsilon_h^{k_i}$ forms a bounded martingale difference sequence. We bound each term on the RHS separately. For the first one, we use the Azuma-Hoeffding inequality which can be found in Lemma B.7. For the second term, we generalize the bound on the summation of the bonus functions over all the episodes from [KSWY21], and show that a similar bound holds for the summation of the bonus over *any* set of episodes $K'$ (see Lemma B.6). Putting everything together, we get that $|K'| = O\left(1/(4^i \mathrm{gap}_{\min}) \cdot H^4 \cdot \log^4 T \cdot \mathrm{poly}(d_{\mathcal{F}})\right)$, where $d_{\mathcal{F}}$ is the complexity parameter of the class.

## 5 Conclusion and Societal Impact

In this paper, we consider episodic RL with general function approximation. We prove that there are algorithms with logarithmic adaptivity complexity both in the model-free and model-based settings that achieve logarithmic instance-dependent regret guarantees. This is theoretical work and does not have any negative societal implications.

## Acknowledgments and Disclosure of Funding

Grigoris Velegkas is supported by NSF (IIS-1845032), an Onassis Foundation PhD Fellowship and a Bodossaki Foundation PhD Fellowship. Amin Karbasi acknowledges funding in direct support of this work from NSF (IIS-1845032), ONR (N00014- 19-1-2406), and the AI Institute for Learning-Enabled Optimization at Scale (TILOS). The authors would like to thank the anonymous reviewers for helpful comments and suggestions.

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
