# A Omitted Details from Main Body

## A.1 Minimum Suboptimality Gap Assumption

It is known that without any instance-specific assumptions we cannot get $o(\sqrt{T})$ regret. The sub-optimality gap assumption is standard in the bandit literature and in the RL-literature that provides $o(\sqrt{T})$ regret guarantees. We remark that $\mathrm{gap}_{\inf} > 0$ for all non-degenerate finite action-state spaces. Notice that $\mathrm{gap}_h(s, a) = V_h^*(s) - Q_h^*(s, a), \mathrm{gap}_{\min} = \inf h, s, a\{\mathrm{gap}_h(s, a) : \mathrm{gap}_h(s, a) \neq 0\}$ (if this set is empty the MDP is degenerate). Notice that if there are multiple optimal actions $a_i$ at some state $s$, then $\mathrm{gap}_h(s, a_i) = 0$, so they will not be considered in the minimization. Thus, the multiplicity of the optimal policies does not break the assumption. Note that $\mathrm{gap}_{\min} = 0$ only in an infinite state-action space such that for all $\varepsilon > 0$, there are some $h_\varepsilon, s_\varepsilon, a_\varepsilon$ such that $0 < V_{h_\varepsilon}^*(s_\varepsilon) - Q_{h_\varepsilon}^*(s_\varepsilon, a_\varepsilon) \leq \varepsilon$ (i.e. we get arbitrarily close to an optimal action). Hence, it quantifies the hardness of the underlying problem because it provides a gap between the reward of an optimal policy and the reward of the non-optimal policies, so it shows how "easy" it is to distinguish an optimal one.

## A.2 Omitted Algorithms

---
**Algorithm 4** Model-Free Sampling Routine

---
**Require:** Function class $\mathcal{F}$, current sub-sampled dataset $\widehat{\mathcal{Z}}$, new element $z = (s, a)$, failure probability $\delta \in (0, 1)$
1: Let $p_z$ be the smallest number such that $1/p_z$ is an integer and $p_z$ is greater than

$$\min\{1, C\mathrm{sensitivity}_{\widehat{\mathcal{Z}}, \mathcal{F}} \log(T\mathcal{N}(\mathcal{F}, \sqrt{\delta/(64T^3)})/\delta)\}$$

2: Let $\widehat{z} \in \mathcal{C}(\mathcal{S} \times \mathcal{A}, 1/16\sqrt{64T^3/\delta})$ such that

$$\sup_{f \in \mathcal{F}} |f(z) - f(\widehat{z})| \leq 1/16\sqrt{64T^3/\delta}$$

3: Add $1/p_z$ copies of $\widehat{z}$ into $\widehat{\mathcal{Z}}$ with probability $p_z$
4: **Return** $\widehat{\mathcal{Z}}$

---

---
**Algorithm 5** Model-Based Sampling Routine

---
**Require:** Function class $\mathcal{F}$, current sub-sampled dataset $\widehat{\mathcal{Z}}$, new element $z = (s, a, V)$, failure probability $\delta \in (0, 1)$
1: Let $p_z$ be the smallest number such that $1/p_z$ is an integer and $p_z$ is greater than

$$\min\{1, C \cdot \mathrm{sensitivity}_{\widehat{\mathcal{Z}}, \mathcal{F}} \cdot \log(T\mathcal{N}(\mathcal{F}, \sqrt{\delta/(64T^3)})/\delta)\}$$

2: Add $1/p_z$ copies of $z$ into $\widehat{\mathcal{Z}}$ with probability $p_z$
3: **Return** $\widehat{\mathcal{Z}}$

---

# B Proof of Theorem 3.1

In this section, our main goal is to prove Theorem 3.1. Recall that we assume we have access to a set $\mathcal{F} \subseteq \{f : \mathcal{S} \times \mathcal{A} \to [0, H + 1]\}$, which we use to approximate the Q-function. For this set, we work with Assumption 2.2 and Assumption 2.3.

The proofs of the supporting lemmas are postponed to Appendix B.1. Before we are ready to prove our result, we need to discuss some results of prior works that are crucial to our proof.

The regret decomposition in [HZG21], gives us that

**Lemma B.1.** *[HZG21] For any MDP $M$ we have that*

$$\mathbb{E}\left[\text{Regret}(K)\right] = \mathbb{E}\left[\sum_{k=1}^{K}\sum_{h=1}^{H}\text{gap}_h(s_h^k, a_h^k)\right].$$

*Moreover, for any $\tau > 0$ it holds with probability at least $1 - \lceil\log T\rceil e^{-\tau}$ that*

$$\text{Regret}(K) \leq 2\sum_{k=1}^{K}\sum_{h=1}^{H}\text{gap}_h(s_h^k, a_h^k) + \frac{16H^2\tau}{3} + 2.$$

The following lemma resembles Lemma 6.3 [HZG21]. Its proof is postponed to Appendix B.1.

**Lemma B.2.** *If we pick*

$$\beta = CH^2\log(T\mathcal{N}(\mathcal{F}, \delta/T^2)/\delta)\dim_E(\mathcal{F}, 1/T)\log^2 T\log\left(\mathcal{C}(\mathcal{S}\times\mathcal{A}, \delta/T^2)T/\delta)\right),$$

*for come constant $C$ and for $h \in [H]$, then we have that with probability at least $1 - 2K\delta$*

$$\sum_{k=1}^{K}\left(V_h^*(s_h^k) - Q_h^*(s_h^k, a_h^k)\right) \leq \frac{CH^4\dim_E^2(\mathcal{F}, 1/T)\log^2 T\log(T\mathcal{N}(\mathcal{F}, \delta/T^2)/\delta)\log(\mathcal{C}(\mathcal{S}\times\mathcal{A}, \delta/T^2)T/\delta)}{\text{gap}_{\min}}.$$

We are now ready to state the regret bound of our algorithm. In particular the regret guarantee follows from the regret decomposition and the bound we established before.

**Lemma B.3.** *There exists a constant $C$ and proper values of the parameter $\beta$ of Algorithm 1 such that with probability at least $1 - \lceil\log T\rceil e^{-\tau} - 2K\delta$ the regret of the algorithm is bounded by*

$$\text{Regret}(K) \leq \frac{CH^5\dim_E^2(\mathcal{F}, 1/T)\log^2 T\log(T\mathcal{N}(\mathcal{F}, \delta/T^2)/\delta)\log(\mathcal{C}(\mathcal{S}\times\mathcal{A}, \delta/T^2)T/\delta)}{\text{gap}_{\min}} + \frac{16H^2\tau}{3} + 2.$$

*The choice of the parameter is*

$$\beta = CH^2\log(T\mathcal{N}(\mathcal{F}, \delta/T^2)/\delta)\dim_E(\mathcal{F}, 1/T)\log^2 T\log\left(\mathcal{C}(\mathcal{S}\times\mathcal{A}, \delta/T^2)T/\delta)\right).$$

*Proof.* Throughout the proof, we condition on the events described in Lemma B.1, B.2 which happen with probability at least $1 - \lceil\log T\rceil e^{-\tau} - 2K\delta$.

From Lemma B.1 we have that

$$\text{Regret}(K) \leq 2\sum_{k=1}^{K}\sum_{h=1}^{H}\text{gap}_h(s_h^k, a_h^k) + \frac{16H^2\tau}{3} + 2.$$

We can bound the first term on the RHS using Lemma B.2 as follows

$$2\sum_{k=1}^{K}\sum_{h=1}^{H}\text{gap}_h(s_h^k, a_h^k) = 2\sum_{k=1}^{K}\sum_{h=1}^{H}\left(V_h^*(s_h^k) - Q_h^*(s_h^k, a_h^k)\right)$$

$$\leq \frac{CH^5\dim_E^2(\mathcal{F}, 1/T)\log^2 T\log(T\mathcal{N}(\mathcal{F}, \delta/T^2)/\delta)\log(\mathcal{C}(\mathcal{S}\times\mathcal{A}, \delta/T^2)T/\delta)}{\text{gap}_{\min}}.$$

Hence, for the total regret we have that

$$\text{Regret}(K) \leq \frac{CH^5\dim_E^2(\mathcal{F}, 1/T)\log^2 T\log(T\mathcal{N}(\mathcal{F}, \delta/T^2)/\delta)\log(\mathcal{C}(\mathcal{S}\times\mathcal{A}, \delta/T^2)T/\delta)}{\text{gap}_{\min}} + \frac{16H^2\tau}{3} + 2.$$

$\square$

We remark that the value of $\beta$ we use in the main body is simply an upper bound on this value.

Finally, we state the bound on the adaptivity of our algorithm. In particular, since our algorithm is the same as in [KSWY21], the logarithmic switching cost follows directly from their result.

**Lemma B.4.** *[KSWY21] For any fixed $h \in [H]$, With probability $1 - \delta$, the sub-sampled dataset $\widehat{\mathcal{Z}}_h^k$ changes at most*

$$O\left(\log(T\mathcal{N}(\mathcal{F}, \sqrt{\delta/T^3})/\delta) \dim_E(\mathcal{F}, 1/T) \log^2 T\right)$$

*times.*

We are now ready to prove Theorem 3.1.

*Proof of Theorem 3.1:*

The proof of this theorem follows by using Lemma B.3 and taking a union on the result of Lemma B.4 and setting the error probability accordingly. $\square$

## B.1   Supporting Lemmas: Theorem 3.1

In this section, we present the proof of Lemma B.2.

First, we need to show that the sub-sampled dataset is a good approximation of the original one. To this end, we use Proposition 1 from [KSWY21].

**Proposition B.5.**  *[KSWY21] For any $h, k \in [H] \times [K]$ we let*

$$\underline{b}_h^k(\cdot, \cdot) = \sup_{\|f_1 - f_2\|^2_{\mathcal{Z}_h^k} \leq \beta/100} |f_1(\cdot, \cdot) - f_2(\cdot, \cdot)|,$$

$$\overline{b}_h^k(\cdot, \cdot) = \sup_{\|f_1 - f_2\|^2_{\mathcal{Z}_h^k} \leq 100\beta} |f_1(\cdot, \cdot) - f_2(\cdot, \cdot)|.$$

*Then, with probability at least $1 - \delta/32$ we have that*

$$\underline{b}_h^k(\cdot, \cdot) \leq b_h^k(\cdot, \cdot) \leq \overline{b}_h^k(\cdot, \cdot).$$

We now present a generalized version of Lemma 11 [KSWY21] that will be used to bound the regret of our algorithm. Essentially, this gives a bound on the summation of the bonus terms over a set of episodes $K' \subseteq [K]$ in terms of the eluder dimension of the function class and the number of episodes.

**Lemma B.6.** *For any set $K' \subseteq [K]$, with probability at least $1 - \delta/32$ we have that*

$$\sum_{i=1}^{|K'|} \sum_{h=1}^{H} b_h^{k_i}(s_h^{k_i}, a_h^{k_i}) \leq H + H(H+1) \cdot \dim_E(\mathcal{F}, 1/T) + CH\sqrt{\dim_E(\mathcal{F}, 1/T)|K'| \cdot \beta},$$

*where $C > 0$ is some constant.*

*Proof.* Throughout the proof, we condition on the event defined in Proposition B.5. This gives us that for any $k \in K', h \in H$

$$b_h^k(s_h^k, a_h^k) \leq \overline{b}_h^k(s_h^k, a_h^k) = \sup_{\|f_1 - f_2\|^2_{\mathcal{Z}_h^k} \leq 100\beta} |f_1(s_h^k, a_h^k) - f_2(s_h^k, a_h^k)|.$$

We bound $\sum_{i=1}^{K'} \overline{b}_h^{k_i}(s_h^{k_i}, a_h^{k_i})$ for each $h \in [H]$ separately.

Given some $\epsilon > 0$, we define $\mathcal{L}_h = \{(s_h^{k_i}, a_h^{k_i}) : k_i \in K', \overline{b}_h^{k_i}(s_h^{k_i}, a_h^{k_i}) > \epsilon\}$, i.e. the set of state-action pairs at step $h$ and some episode in $K'$ where the bonus function has value greater than $\epsilon$. Consider some $k \in K'$. We denote $L_h = |\mathcal{L}_h|$, $\widetilde{\mathcal{Z}}_h^k = \{(s_h^k, a_h^k) \in \mathcal{Z}_h^k, k \in K'\}$. Our goal is to show that there is some $z_h^k = (s_h^k, a_h^k) \in \mathcal{L}_h$ that is $\epsilon$-dependent on at least $L_h/\dim_E(\mathcal{F}, \epsilon) - 1$ disjoint subsequences in $\mathcal{Z}_h^k \cap \mathcal{L}_h$. We also denote $N = L_h/\dim_E(\mathcal{F}, \epsilon) - 1$.

To do that, we decompose $\mathcal{L}_h$ into $N+1$ disjoint subsets and we denote the $j$-th subset by $\mathcal{L}_h^j$. We use the following procedure. Initially we set $\mathcal{L}_h^j = \emptyset$ for all $j \in [N+1]$ and consider every $z_h^k \in \mathcal{L}_h$ in a sequential manner. For each such $z_h^k$ we find the smallest index $j$, $1 \leq j \leq N$, such that $z_h^k$ is $\epsilon$-independent of the elements in $\mathcal{L}_h^j$ with respect to $\mathcal{F}$. If there is no such $j$, we set $j = N+1$. Then, we update $\mathcal{L}_h^j \leftarrow \mathcal{L}_h^j \cup z_h^k$. Notice that after we go through all the elements of $\mathcal{L}_h$, we must have that $\mathcal{L}_h^{N+1} \neq \emptyset$. This is because every set $\mathcal{L}_h^j, 1 \leq j \leq N$, contains at most $\dim_E(\mathcal{F}, \epsilon)$ elements. Moreover, by definition, every element $z_h^k \in \mathcal{L}_h^{N+1}$ is $\epsilon$-dependent on $N = L_h/\dim_E(\mathcal{F}, \epsilon) - 1$ disjoint subsequences in $\mathcal{Z}_h^k \cap \mathcal{L}^h$.

Furthermore, since $\bar{b}_h^k(s_h^k, a_h^k) > \epsilon$ for all $z_h^k \in \mathcal{L}_h$ there must exist $f_1, f_2 \in \mathcal{F}$ such that $|f_1(s_h^k, a_h^k) - f_2(s_h^k, a_h^k)| > \epsilon$ and $||f_1 - f_2||_{\mathcal{Z}_h^k}^2 \leq 100\beta$. Hence, since $z_h^k \in \mathcal{L}_h^{N+1}$ is $\epsilon$-dependent on $N$ disjoint subsequences $\mathcal{L}_h^j$ and for each such subsequence, by the definition of $\epsilon$-dependence, it holds that $||f_1 - f_2||_{\mathcal{L}_h^j} > \epsilon^2$ we have that

$$N\epsilon^2 \leq ||f_1 - f_2||_{\mathcal{Z}_h^k}^2 \leq 100\beta \implies (L_h/\dim_E(\mathcal{F}, \epsilon) - 1)\epsilon^2 \leq 100\beta \implies L_h \leq \left(\frac{100\beta}{\epsilon^2} + 1\right)\dim_E(\mathcal{F}, \epsilon).$$

We now pick a permutation $\bar{b}_1 \geq \bar{b}_2 \geq \ldots \geq \bar{b}_{|K'|}$ of the bonus functions $\{\bar{b}_h^k(s_h^k, a_h^k)\}_{k \in K'}$. For all $\bar{b}_k \geq 1/|K'|$ it holds that

$$k \leq \left(\frac{100\beta}{\bar{b}_k^2} + 1\right)\dim_E(\mathcal{F}, \bar{b}_k) \leq \left(\frac{100\beta}{\bar{b}_k^2} + 1\right)\dim_E(\mathcal{F}, 1/K') \implies$$

$$\bar{b}_k \leq \left(\frac{k}{\dim_E(\mathcal{F}, 1/K')} - 1\right)^{-1/2}\sqrt{100\beta}.$$

Moreover, notice that we get by definition that $\bar{b}_k \leq H + 1$. Hence, we have that

$$\begin{aligned}
\sum_{i=1}^{|K'|} \bar{b}_h^{k_i}(s_h^{k_i}, a_h^{k_i}) &= \sum_{i:\bar{b}_{k_i} < 1/|K'|} \bar{b}_h^{k_i}(s_h^{k_i}, a_h^{k_i}) + \sum_{i:\bar{b}_{k_i} \geq 1/|K'|} \bar{b}_h^{k_i}(s_h^{k_i}, a_h^{k_i}) \\
&\leq |K'| \cdot 1/|K'| + \sum_{i:\bar{b}_{k_i} \geq 1/|K'|, i \leq \dim_E(\mathcal{F}, 1/|K'|)} \bar{b}_h^{k_i}(s_h^{k_i}, a_h^{k_i}) \\
&\quad + \sum_{i:\bar{b}_{k_i} \geq 1/|K'|, |K'| \geq i > \dim_E(\mathcal{F}, 1/|K'|)} \bar{b}_h^{k_i}(s_h^{k_i}, a_h^{k_i}) \\
&\leq 1 + (H+1) \cdot \dim_E(\mathcal{F}, 1/|K'|) + \sum_{|K'| \geq i > \dim_E(\mathcal{F}, 1/|K'|)} \left(\frac{k}{\dim_E(\mathcal{F}, 1/|K'|)} - 1\right)^{-1/2}\sqrt{100\beta} \\
&\leq 1 + (H+1) \cdot \dim_E(\mathcal{F}, 1/|K'|) + C\sqrt{\dim_E(\mathcal{F}, 1/|K'|)|K'|\beta} \\
&\leq 1 + (H+1) \cdot \dim_E(\mathcal{F}, 1/T) + C\sqrt{\dim_E(\mathcal{F}, 1/T)|K'|\beta}
\end{aligned}$$

for some constant $C > 0$, where the second to last inequality can be obtained by bounding the summation by the integral and the last one by the definition of the eluder dimension. Summing up all the inequalities for $h \in [H]$, we get the result. $\qquad\square$

We need the Azuma-Hoeffding inequality to bound a martingale difference sequence. For completeness, we present it here as well.

**Lemma B.7.** *[CBL06] Let $\{x_i\}_{i=1}^n$ be a martingale difference sequence with respect to some filtration $\{\mathcal{F}_i\}$ for which $|x_i| \leq M$ for some constant $M$, $x_i$ is $\mathcal{F}_{i+1}$ measurable and $\mathbb{E}[x_i|\mathcal{F}_i] = 0$.*

*Then, for any $0 < \delta < 1$, we have that with probability at least $1 - \delta$ it holds that*

$$\sum_{i=1}^{n} x_i \leq M\sqrt{2n\log(1/\delta)}.$$

The following lemma that appears in [KSWY21] shows that the estimate of the $Q$-function upper bounds the optimal one.

**Lemma B.8.** *[KSWY21] With probability at least $1 - \delta/2$ we have that for all $(k, h) \in [K] \times [H]$ and all $(s, a) \in \mathcal{S} \times \mathcal{A}$*

$$Q_h^*(s, a) \leq Q_h^k(s, a) \leq \bar{f}_h^k(s, a) + 2b_h^k(s, a)$$

*where $\bar{f}_h^k(\cdot, \cdot) = \sum_{s' \in \mathcal{S}} P_h(s'|\cdot, \cdot)V_{h+1}^k(s') + r_h(\cdot, \cdot)$.*

We now present a lemma that bounds the number of rounds that the suboptimilaty gap falls in some interval. It is inspired by Lemma 6.2 [HZG21].

**Lemma B.9.** *If we pick*

$$\beta = CH^2 \log(T\mathcal{N}(\mathcal{F}, \delta/T^2)/\delta) \dim_E(\mathcal{F}, 1/T) \log^2 T \log\left(\mathcal{C}(\mathcal{S} \times \mathcal{A}, \delta/T^2)T/\delta\right),$$

*then there exists a constant $\widetilde{C}$ such that for all $h \in [H], n \in [N]$ with probability at least $1 - 2K\delta$, we have that*

$$\sum_{k=1}^{K} \mathbb{1}[V_h^*(s_h^k) - Q_h^{\pi_k}(s_h^k, a_h^k) \geq 2^n\mathrm{gap}_{\min}] \leq \frac{\widetilde{C}H^4\dim_E^2(\mathcal{F}, 1/T)\log^2 T \log(T\mathcal{N}(\mathcal{F}, \delta/T^2)/\delta)\log(\mathcal{C}(\mathcal{S} \times \mathcal{A}, \delta/T^2)T/\delta)}{4^n\mathrm{gap}_{\min}^2}.$$

*Proof.* We keep $h$ fixed.

We denote by $K'$ the set of episodes where the gap at step $h$ is at least $2^n$, i.e.

$$K' = \left\{k \in [K] : V_h^*(s_h^k) - Q_h^{\pi_k}(s_h^k, a_h^k) \geq 2^n\mathrm{gap}_{\min}\right\}.$$

The goal is to bound the quantity $\sum_{i=1}^{|K'|}\left(Q_h^{k_i}(s_h^{k_i}, a_h^{k_i}) - Q_h^{\pi_{k_i}}(s_h^{k_i}, a_h^{k_i})\right)$ from below and above with functions $f_1(|K'|), f_2(|K'|)$ and then use the fact that $f_1(|K'|) \leq f_2(|K'|)$ to derive an upper bound on $|K'|$.

For the lower bound, we have that

$$\sum_{i=1}^{|K'|}\left(Q_h^{k_i}(s_h^{k_i}, a_h^{k_i}) - Q_h^{\pi_{k_i}}(s_h^{k_i}, a_h^{k_i})\right) \geq \sum_{i=1}^{|K'|}\left(Q_h^{k_i}(s_h^{k_i}, \pi_h^*(s_h^{k_i})) - Q_h^{\pi_{k_i}}(s_h^{k_i}, a_h^{k_i})\right)$$

$$\geq \sum_{i=1}^{|K'|}\left(Q_h^*(s_h^{k_i}, \pi_h^*(s_h^{k_i})) - Q_h^{\pi_{k_i}}(s_h^{k_i}, a_h^{k_i})\right)$$

$$= \sum_{i=1}^{|K'|}\left(V_h^*(s_h^{k_i}) - Q_h^{\pi_{k_i}}(s_h^{k_i}, a_h^{k_i})\right) \geq 2^n\mathrm{gap}_{\min}|K'|,$$

where the first inequality holds by the definition of the policy $\pi_{k_i}$, the second one follows because $Q_h^{k_i}(\cdot, \cdot)$ is an optimistic estimate of $Q_h^*(\cdot, \cdot)$ which happens with probability at least $1 - \delta/2$ (see Lemma B.8) and the third one by the definition of $k_i$.

We get the upper bound on this quantity in the following way. For any $h' \in [H]$ we have

$$Q_{h'}^k(s_{h'}^k, a_{h'}^k) - Q_{h'}^{\pi_k}(s_{h'}^k, a_{h'}^k) \leq \sum_{s' \in \mathcal{S}} P_{h'}(s'|s_{h'}^k, a_{h'}^k)V_{h'+1}^k(s') + r_{h'}(s_{h'}^k, a_{h'}^k) + 2b_{h'}^k(s_{h'}^k, a_{h'}^k) - Q_{h'}^{\pi_k}(s_{h'}^k, a_{h'}^k)$$

$$= \left\langle P_{h'}(\cdot|s_{h'}^k, a_{h'}^k), V_{h'+1}^k - V_{h'+1}^{\pi_k}\right\rangle + 2b_{h'}^k(s_{h'}^k, a_{h'}^k)$$

$$= V_{h'+1}^k(s_{h'+1}^k) - V_{h'+1}^{\pi_k}(s_{h'+1}^k) + \epsilon_{h'}^k + 2b_{h'}^k(s_{h'}^k, a_{h'}^k)$$

$$= Q_{h'+1}^k(s_{h'+1}^k, a_{h'+1}^k) - Q_{h'+1}^{\pi_k}(s_{h'+1}^k, a_{h'+1}^k) + \epsilon_{h'}^k + 2b_{h'}^k(s_{h'}^k, a_{h'}^k)$$

where we define $\epsilon_{h'}^k = \left\langle P_{h'}(\cdot|s_{h'}^k, a_{h'}^k), V_{h'+1}^k - V_{h'+1}^{\pi_k} \right\rangle - (V_{h'+1}^k(s_{h'+1}^k) - V_{h'+1}^{\pi_k}(s_{h'+1}^k))$ and the inequality follows from Lemma B.8.

We now take the summation over all $k \in |K'|, h \leq h' \leq H$ and we get

$$\sum_{i=1}^{|K'|} \left( Q_h^{k_i}(s_h^{k_i}, a_h^{k_i}) - Q_h^{\pi_{k_i}}(s_h^{k_i}, a_h^{k_i}) \right) \leq \sum_{i=1}^{|K'|} \sum_{h'=h}^H \epsilon_{h'}^{k_i} + \sum_{i=1}^{|K'|} \sum_{h'=h}^H b_{h'}^{k_i}(s_{h'}^{k_i}, a_{h'}^{k_i}).$$

We will bound each of the two terms on the RHS separately.

For the first term, we notice that $x_j = \left\langle P_j(\cdot|s_j^{k_i}, a_j^{k_i}), V_{j+1}^{k_i} - V_{j+1}^{\pi_{k_i}} \right\rangle - (V_{j+1}^{k_i}(s_{j+1}^{k_i}) - V_{j+1}^{\pi_{k_i}}(s_{j+1}^{k_i}))$ forms a martingale difference sequence with zero mean and $|x_j| \leq 2H$. Hence, we can use Lemma B.7 and that for each $k \in K'$, with probability at least $1 - \delta$ we have that

$$\sum_{i=1}^k \sum_{j=1}^H \left( \left\langle P_j(\cdot|s_j^{k_i}, a_j^{k_i}), V_{j+1}^{k_i} - V_{j+1}^{\pi_{k_i}} \right\rangle - \left( V_{j+1}^{k_i}(s_{j+1}^{k_i}) - V_{j+1}^{\pi_{k_i}}(s_{j+1}^{k_i}) \right) \right) \leq \sqrt{8kH^3 \log(1/\delta)}.$$

If we take the union bound over all $k \in [K]$ we have that with probability at least $1 - |K'|\delta$

$$\sum_{i=1}^{|K'|} \sum_{h'=h}^H \epsilon_{h'}^{k_i} \leq \sqrt{8|K'|H^3 \log(1/\delta)}.$$

We now focus on the second term. Using Lemma B.6 we get that

$$\sum_{i=1}^{|K'|} \sum_{h'=h}^H b_{h'}^{k_i}(s_{h'}^{k_i}, a_{h'}^{k_i}) \leq H + H(H+1)\dim_E(\mathcal{F}, 1/T) + CH\sqrt{\dim_E(\mathcal{F}, 1/T)|K'|\beta}$$

and this happens with probability at least $1 - \delta/32$. Hence, combining the upper and lower bound of

$$\sum_{i=1}^{|K'|} \left( Q_h^{k_i}(s_h^{k_i}, a_h^{k_i}) - Q_h^{\pi_{k_i}}(s_h^{k_i}, a_h^{k_i}) \right)$$

we get that

$$2^n \text{gap}_{\min}|K'| \leq \sqrt{8|K'|H^3 \log(1/\delta)} + H + H(H+1)\dim_E(\mathcal{F}, 1/T) + CH\sqrt{\dim_E(\mathcal{F}, 1/T)|K'|\beta}.$$

Solving for $|K'|$ gives us that

$$|K'| \leq \frac{\widetilde{C}H^4 \dim_E^2(\mathcal{F}, 1/T)\log^2 T \log(T\mathcal{N}(\mathcal{F}, \delta/T^2)/\delta)\log(\mathcal{C}(\mathcal{S} \times \mathcal{A}, \delta/T^2)T/\delta)}{4^n \text{gap}_{\min}^2}.$$

$\square$

We are now ready to prove Lemma B.2.

*Proof of Lemma B.2*: Throughout this proof we condition on the event described in Lemma B.9 which happens with probability at least $1 - 2K\delta$. Since $\text{gap}_{\min} > 0$ whenever we do not take the optimal action, we have that either $V_h^*(s_k) - Q_h^*(s_h^k, a_h^k) = 0$ or $V_h^*(s_k) - Q_h^*(s_h^k, a_h^k) \geq \text{gap}_{\min}$. Our approach is to divide the interval $[0, H]$ into $N = \lceil \log(H/\text{gap}_{\min}) \rceil$ intervals and count the number

of $V_h^*(s_k) - Q_h^*(s_h^k, a_h^k)$ that fall into each interval. Notice that for every $V_h^*(s_k) - Q_h^*(s_h^k, a_h^k)$ that falls into interval $i$ we can get an upper bound of $V_h^*(s_k) - Q_h^*(s_h^k, a_h^k) \leq 2^i \text{gap}_{\min}$ and this upper bound is essentially tight. Hence, we have that

$$
\begin{aligned}
\sum_{k=1}^{K} \left( V_h^*(s_h^k) - Q_h^*(s_h^k, a_h^k) \right) &\leq \sum_{i=1}^{N} \sum_{k=1}^{K} \mathbb{1}\left[ 2^i \text{gap}_{\min} \geq V_h^*(s_h^k) - Q_h^*(s_h^k, a_h^k) \geq 2^{i-1} \text{gap}_{\min} \right] \cdot 2^i \text{gap}_{\min} \\
&\leq \sum_{i=1}^{N} \sum_{k=1}^{K} \mathbb{1}\left[ V_h^*(s_h^k) - Q_h^*(s_h^k, a_h^k) \geq 2^{i-1} \text{gap}_{\min} \right] \cdot 2^i \text{gap}_{\min} \\
&\leq \sum_{i=1}^{N} \frac{\widetilde{C} H^4 \dim_E^2(\mathcal{F}, 1/T) \log^2 T \log(T\mathcal{N}(\mathcal{F}, \delta/T^2)/\delta) \log(\mathcal{C}(\mathcal{S} \times \mathcal{A}, \delta/T^2)T/\delta)}{4^{i-1} \text{gap}_{\min}^2} \cdot 2^i \text{gap}_{\min} \\
&= \sum_{i=1}^{N} \frac{\widetilde{C} H^4 \dim_E^2(\mathcal{F}, 1/T) \log^2 T \log(T\mathcal{N}(\mathcal{F}, \delta/T^2)/\delta) \log(\mathcal{C}(\mathcal{S} \times \mathcal{A}, \delta/T^2)T/\delta)}{2^i \text{gap}_{\min}} \\
&\leq \frac{C H^4 \dim_E^2(\mathcal{F}, 1/T) \log^2 T \log(T\mathcal{N}(\mathcal{F}, \delta/T^2)/\delta) \log(\mathcal{C}(\mathcal{S} \times \mathcal{A}, \delta/T^2)T/\delta)}{\text{gap}_{\min}}
\end{aligned}
$$

where the first inequality holds by the definition of the intervals, the second due to the properties of the indicator function, the third because of Lemma B.9 and in the last two steps we just manipulate the constants. $\square$

## C Proof of Theorem 3.2

In this section, our main goal is to prove Theorem 3.2. We work with Assumption 2.4. We follow the same regret decomposition as in Appendix B.

We first present a lemma that is crucial in bounding the regret of the algorithm.

**Lemma C.1.** *If we pick*

$$
\beta = 4H^2 \log(2\mathcal{N}(\mathcal{F}, 1/T)/\delta) + 4/H \left( C + \sqrt{H^2/4 \log(4(K(K+1)/\delta))} \right),
$$

*for come constant C and for $h \in [H]$, then we have that with probability at least $1 - (K+3)\delta$*

$$
\sum_{k=1}^{K} \left( V_h^*(s_h^k) - Q_h^*(s_h^k, a_h^k) \right) \leq \frac{C H^4 \log(T\mathcal{N}(\mathcal{F}, 1/T)/\delta) \dim_E^2(\mathcal{F}, 1/T)}{\text{gap}_{\min}}.
$$

We are now ready to state the regret bound of our algorithm.

**Lemma C.2.** *There exists a constant C and proper values of the parameter $\beta$ of Algorithm 1 such that with probability at least $1 - \lceil \log T \rceil e^{-\tau} - H(K+3)\delta$ the regret of the algorithm is bounded by*

$$
\text{Regret}(K) \leq \frac{C H^5 \log(T\mathcal{N}(\mathcal{F}, 1/T)/\delta) \dim_E^2(\mathcal{F}, 1/T)}{\text{gap}_{\min}} + \frac{16 H^2 \tau}{3} + 2.
$$

*The value of the parameter is*

$$
\beta = 4H^2 \log(2\mathcal{N}(\mathcal{F}, 1/T)/\delta) + 4/H \left( C + \sqrt{H^2/4 \log(4(K(K+1)/\delta))} \right).
$$

*where $\mathcal{N}(\mathcal{F}, 1/T) = \arg\max_{h \in [H]} \mathcal{N}(\mathcal{F}_h, 1/T)$.*

*In particular, the dependence of the regret in the time horizon $T$ is logarithmic.*

*Proof.* Throughout the proof, we condition on the events described in Lemma B.1, C.1 which happen with probability at least $1 - \lceil \log T \rceil e^{-\tau} - H(K+3)\delta$.

From Lemma B.1 we have that

$$\text{Regret}(K) \leq 2 \sum_{k=1}^{K} \sum_{h=1}^{H} \text{gap}_h(s_h^k, a_h^k) + \frac{16H^2\tau}{3} + 2.$$

We can bound the first term on the RHS using Lemma C.1 as follows

$$2 \sum_{k=1}^{K} \sum_{h=1}^{H} \text{gap}_h(s_h^k, a_h^k) = 2 \sum_{k=1}^{K} \sum_{h=1}^{H} \left( V_h^*(s_h^k) - Q_h^*(s_h^k, a_h^k) \right) \leq \frac{2CH^5 \log(T\mathcal{N}(\mathcal{F}, 1/T)/\delta) \dim_E^2(\mathcal{F}, 1/T)}{\text{gap}_{\min}}.$$

This gives us the result.

$\square$

We remark that the value of $\beta$ we use in the main body is simply an upper bound on this value.

The only thing that we need to do now is to bound the number of rounds that we update our policy. Since we are using exactly the same sensitivity score and update probability as in [KSWY21], this follows from their result.

**Lemma C.3.** *[KSWY21] With probability at least $1 - \delta/32$ for any fixed $h \in [H]$ we have that the sub-sampled dataset $\widehat{Z}_h^k, k \in [K]$ changes at most*

$$S_{\max} = C \cdot \log(T\mathcal{N}(F_h, \sqrt{\delta/(64T^3)})/\delta) \dim_E(\mathcal{F}_h, 1/T) \log^2 T$$

*times.*

We are now ready to prove Theorem 3.2.

*Proof of Theorem 3.2:*

=The proof of this theorem follows by combining Lemma C.2 and taking a union on the result of Lemma C.3 and setting the error probability accordingly. $\square$

## C.1 Supporting Lemmas: Theorem 3.2

In this section our goal is to prove the supporting lemmas of Theorem 3.2.

Recall that our approach is to modify the algorithm in [KSWY21] to work in this setting and use a similar analysis as in Appendix B. Unlike Appendix B where we approximate the optimal Q-function, here we try to estimate the true transition kernel. Let $\mathcal{Z}_h^k = \{(s_h^\tau, a_h^\tau, V_{h+1}^\tau(\cdot))\}_{\tau \in [k-1]}$ be the dataset up to episode $k$ and $\widehat{\mathcal{Z}}_h^k$ the sub-sampled dataset. In each episode $k$, we update our policy whenever we add an element in the dataset for some $h \in [H]$. Recall that whenever we perform an update our policy becomes:

$$
\begin{aligned}
Q_{H+1}^k(s, a) &= 0, \\
V_{H+1}^k(s) &= 0, \\
Q_h^k(s, a) &= \min\{r_h(s, a) + \langle \widehat{P}_h^k(\cdot|s, a), V_{h+1}^k \rangle + b_h^k(s, a), H\}, \\
V_h^k(s) &= \max_{a \in \mathcal{A}} Q_h^k(s, a)
\end{aligned}
$$

for some $\widehat{P}_h^k, b_h^k(\cdot, \cdot)$ that we will define shortly. We get the policy $\pi_h^k(s)$ by picking greedily the action that maximizes the estimate $Q_h^k(s, a)$.

The least-squares estimate of the model is

$$\widehat{P}_h^{k+1} = \arg\min_{P \in \mathcal{P}_h} \sum_{k'=1}^{k} \left( \langle P(\cdot|s_h^{k'}, a_h^{k'}), V_{h+1}^{k'} \rangle - y_h^{k'} \right)^2, y_h^{k'} = V_{h+1}^{k'}(s_{h+1}^{k'}).$$

Recall the definition of the function class that we use in the derivation of our results.

**Definition C.4.** Let $\mathcal{V}$ be the set all measurable functions that are bounded by $H$. We now let $f : \mathcal{S} \times \mathcal{A} \times \mathcal{V} \to \mathbb{R}$ and define the following set:

$$\mathcal{F}_h = \left\{ f : \exists P_h \in \mathcal{P}_h \text{ so that } f(s,a,V) = \int_{\mathcal{S}} P_h(s'|s,a)V(s')ds', \forall (s,a,V) \in \mathcal{S} \times \mathcal{A} \times \mathcal{V} \right\}.$$

(1)

Recall that in this setting the norm of a function with respect to a dataset $\mathcal{Z}$ is

$$||f||_{\mathcal{Z}} = \sqrt{\sum_{z=(s_z,a_z,V_z(\cdot))\in\mathcal{Z}} (f(s_z, a_z, V_z(\cdot))^2}.$$

Recall also that the bonus function is

$$b_h^k(s,a) = \sup_{f_1,f_2\in\mathcal{F}_h:\min\{||f_1-f_2||_{\widehat{\mathcal{Z}}_h^k},T(H+1)^2\}\leq\beta} |f_1(s,a,V_{h+1}^k(\cdot)) - f_2(s,a,V_{h+1}^k(\cdot))|.$$

The parameter $\beta$ will be defined later in a way that will ensure optimism.

First, we need to show that at for every $k \in [K], h \in [H]$, the sub-sampled dataset approximates the original one. Our approach is inspired by [KSWY21].

We define the following quantities

$$\underline{\mathcal{C}}_h^k(\alpha) = \left\{ (f_1, f_2) \in \mathcal{F}_h \times \mathcal{F}_h : ||f_1 - f_2||_{\mathcal{Z}_h^k}^2 \leq \alpha/100 \right\}$$

$$\widehat{\mathcal{C}}_h^k(\alpha) = \left\{ (f_1, f_2) \in \mathcal{F}_h \times \mathcal{F}_h : \min\{||f_1 - f_2||_{\widehat{\mathcal{Z}}_h^k}^2, T(H+1)^2\} \leq \alpha \right\}$$

$$\overline{\mathcal{C}}_h^k(\alpha) = \left\{ (f_1, f_2) \in \mathcal{F}_h \times \mathcal{F}_h : ||f_1 - f_2||_{\mathcal{Z}_h^k}^2 \leq 100\alpha \right\}.$$

We also let

$$\underline{b}_h^k(s,a) = \sup_{f_1,f_2\in\underline{\mathcal{C}}_h^k(\beta)} |f_1(s,a,V_{h+1}^k(\cdot)) - f_2(s,a,V_{h+1}^k(\cdot))|$$

$$\overline{b}_h^k(s,a) = \sup_{f_1,f_2\in\overline{\mathcal{C}}_h^k(\beta)} |f_1(s,a,V_{h+1}^k(\cdot)) - f_2(s,a,V_{h+1}^k(\cdot))|.$$

Our goal is to show that $\underline{\mathcal{C}}_h^k(\alpha) \subseteq \widehat{\mathcal{C}}_h^k(\alpha) \subseteq \overline{\mathcal{C}}_h^k(\alpha)$ with high probability. Let $\mathcal{E}_h^k(\alpha)$ denote the event that this holds. We also denote by $\mathcal{E}_h^k = \cap_{n=0}^{\infty}\mathcal{E}_h^k(100^n\beta)$. This event will show us that $\widehat{\mathcal{Z}}_h^k$ is a good approximation to $\mathcal{Z}_h^k$.

Notice that whenever this happens, it holds that $\underline{b}_h^k(s,a) \leq b_h^k(s,a) \leq \overline{b}_h^k(s,a)$.

The following lemma which is inspired by [KSWY21] establishes that fact.

**Lemma C.5.** *The probability that all the events $\mathcal{E}_h^k$ happen satisfies*

$$\Pr\left( \bigcap_{k=1}^{K} \bigcap_{h=1}^{H} \mathcal{E}_h^k \right) \geq 1 - \delta.$$

To prove Lemma C.5 we need the following concentration inequality proved in [Fre75].

**Lemma C.6.** *Let $\{Y_i\}_{i\in\mathbb{N}}$ be a real-valued martingale with difference sequence $\{X_i\}_{i\in\mathbb{N}}$. Let $R$ be a uniform bound on $X_i$. Fix some $n \in \mathbb{N}$ and let $\sigma^2$ be a number such that*

$$\sum_{i=1}^{n} \mathbb{E}[X_i^2|Y_0,\ldots,Y_{i-1}] \leq \sigma^2.$$

*Then, for all $t \geq 0$ we have that*

$$\Pr(|Y_n - Y_0| \geq t) \leq 2\exp\left\{ -\frac{t^2/2}{\sigma^2 + Rt/3} \right\}.$$

Moreover, we need a bound on the number of elements that are in the sub-sampled dataset. This is established in [KSWY21].

**Lemma C.7.** *[KSWY21] We have that with probability at least $1 - \delta/(64T)$, we have $|\widehat{\mathcal{Z}}_h^k| \leq 64T^3/\delta$ for all $\delta > 0$.*

The subsequent lemma shows that, indeed, whenever $\mathcal{E}_h^k$ happens the sub-sampled dataset is a good approximation of the original one. It was proved in [KSWY21].

**Lemma C.8.** *[KSWY21] Whenever the event $\mathcal{E}_h^k$ happens, it holds that*

$$\frac{1}{10000}\|f_1 - f_2\|_{\mathcal{Z}_h^k}^2 \leq \min\{\|f_1 - f_2\|_{\widehat{\mathcal{Z}}_h^k}^2, T(H+1)^2\} \leq 10000\|f_1 - f_2\|_{\mathcal{Z}_h^k}^2, \text{ if } \|f_1 - f_2\|_{\mathcal{Z}_h^k}^2 > 100\beta$$

*and*

$$\min\{\|f_1 - f_2\|_{\widehat{\mathcal{Z}}_h^k}^2, T(H+1)^2\} \leq 10000\beta, \text{ if } \|f_1 - f_2\|_{\mathcal{Z}_h^k}^2 \leq 100\beta.$$

To establish our result, we need the following lemma. The proof follows the approach of [KSWY21]. We present it here for completeness.

**Lemma C.9.** *For any $\alpha \in [\beta, T(H+1)^2]$, a fixed $h \in [H]$ and $k \in [K]$ we have the following bound for the probability that all the events $\{\mathcal{E}_h^i\}_{i \leq k-1}$ happen and the last one does not happen*

$$\Pr\left(\left(\bigcap_{i=1}^{k-1} \mathcal{E}_h^i\right) \mathcal{E}_h^k(\alpha)^c\right) \leq \delta/(32T^2).$$

*Proof.* Let $C_1$ be the quantity the sensitivity in the sampling probability. We fix some $h \in [H]$ throughout the proof.

We consider a fixed pair of functions $f_h^1, f_h^2$ in the discretized set $\mathcal{C}(\mathcal{F}_h, \sqrt{\delta/(64T^3)})$ and for $i \geq 2$ we let

$$Z_i = \max\left\{\|f_h^1 - f_h^2\|_{\mathcal{Z}_h^i}^2, \min\{\|f_h^1 - f_h^2\|_{\widehat{\mathcal{Z}}_h^{i-1}}^2, T(H+1)^2\}\right\}.$$

We also define

$$Y_i = \begin{cases} \frac{1}{p_{z_h^{i-1}}}(f_h^1(z_h^{i-1}) - f_h^2(z_h^{i-1}))^2 & z_h^{i-1} \text{ is added to } \widehat{\mathcal{Z}}_h^i \text{ and } Z_i \leq 2000000\alpha \\ 0, & z_h^{i-1} \text{ is not added to } \widehat{\mathcal{Z}}_h^i \text{ and } Z_i \leq 2000000\alpha \\ (f_h^1(z_h^{i-1}) - f_h^2(z_h^{i-1}))^2 & \text{otherwise} \end{cases}$$

Let $\mathbb{F}_i$ be the filtration that $Y_i$ is adapted to. Our goal is to use Freedman's inequality (i.e. Lemma C.6) for $Y_i$. Notice that $\mathbb{E}[Y_i|\mathbb{F}_i] = (f_h^1(z_h^{i-1}) - f_h^2(z_h^{i-1}))^2$. Now we focus on the variance of $Y_i$. Notice that if $p_{z_h^{i-1}} = 1$ or $Z_i > 2000000\alpha$ then $Y_i$ is deterministic so $Y_i - \mathbb{E}[Y_i|\mathbb{F}_{i-1}] = \text{Var}[Y_i - \mathbb{E}[Y_i|\mathbb{F}_{i-1}]] = 0$. For the other case, recall that

$$p_{z_h^i} = \min\{1, C' \cdot \text{sensitivity}_{\widehat{\mathcal{Z}}_h^{i-1}, \mathcal{F}_h}(z_h^i) \cdot \log(T\mathcal{N}(\mathcal{F}, \sqrt{\delta/64T^3})/\delta)\} = \min\{1, C_1 \cdot \text{sensitivity}_{\widehat{\mathcal{Z}}_h^{i-1}, \mathcal{F}_h}(z_h^i)\}$$

and

$$\text{sensitivity}_{\mathcal{Z}, \mathcal{F}}(z) = \min\left\{\sup_{f_1, f_2 \in \mathcal{F}} \frac{(f_1(z) - f_2(z))^2}{\min\{\|f_1 - f_2\|_{\mathcal{Z}}, T(H+1)^2\} + \beta}, 1\right\}$$

Since $p_{z_h^{i-1}} < 1 \implies C_1 \cdot \text{sensitivity}_{\widehat{\mathcal{Z}}_h^{i-1}, \mathcal{F}_h}(z_h^i) < 1$. We consider two cases. If $Y_i \neq 0$ we can see that $Y_i \geq \mathbb{E}[Y_i|\mathbb{F}_i]$ so $|Y_i - \mathbb{E}[Y_i]| \leq Y_i$. Moreover,

$$
\begin{aligned}
Y_i &\leq \frac{(f_h^1(z_h^{i-1}) - f_h^2(z_h^{i-1}))^2}{p_{z_h^{i-1}}} \\
&\leq \frac{(f_h^1(z_h^{i-1}) - f_h^2(z_h^{i-1}))^2}{C_1 \sup_{f_1, f_2 \in \mathcal{F}_h} \frac{(f_1(z_h^{i-1}) - f_2(z_h^{i-1}))^2}{\min\{\|f_1 - f_2\|_{\widehat{\mathcal{Z}}_h^{i-1}}, T(H+1)^2\} + \beta}} \\
&\leq \frac{(f_h^1(z_h^{i-1}) - f_h^2(z_h^{i-1}))^2 \min\{\|f_1 - f_2\|_{\widehat{\mathcal{Z}}_h^{i-1}}, T(H+1)^2\} + \beta\}}{C_1 (f_h^1(z_h^{i-1}) - f_h^2(z_h^{i-1}))^2} \\
&= \left( \min\{\|f_1 - f_2\|_{\widehat{\mathcal{Z}}_h^{i-1}}, T(H+1)^2\} + \beta \right) \cdot 1/C_1 \\
&\leq 2000001\alpha/C_1 < 3000000\alpha/C_1
\end{aligned}
$$

Thus, we see that $|Y_i - \mathbb{E}[Y_i]| \leq 3000000\alpha/C_1$. On the other hand, we can see that if $Y_i = 0$ then $|\mathbb{E}_{i-1}[Y_i] - Y_i| = (f_h^1(z_h^{i-1}) - f_h^2(z_h^{i-1}))^2$ and the inequality we derived above still holds.

For the variance, we can see that

$$
\begin{aligned}
\text{Var}[Y_i - \mathbb{E}[Y_i|\mathbb{F}_i]|\mathbb{F}_i] &= p_{z_h}^{i-1} \left( \frac{1}{p_{z_h^{i-1}}} (f_h^1(z_h^{i-1}) - f_h^2(z_h^{i-1}))^2 \right)^2 + (1 - p_{z_h}^{i-1}) \cdot 0 \\
&\leq \frac{1}{p_{z_h^{i-1}}} (f_h^1(z_h^{i-1}) - f_h^2(z_h^{i-1}))^4 \\
&\leq 3000000\alpha \left( f_h^1(z_h^{i-1}) - f_h^2(z_h^{i-1}) \right)^2 /C_1
\end{aligned}
$$

where the first equality follows from the definition, the first inequality is trivial and the third one from the inequality we derived above. Let $k'$ be the maximum number $\leq k$ such that $Z_{k'} \leq 2000000\alpha$. Summing up the above inequalities for $i = 2, \ldots, k$ we get

$$
\begin{aligned}
\sum_{i=2}^k \text{Var}[Y_i - \mathbb{E}[Y_i|\mathbb{F}_i]|\mathbb{F}_i] &= \sum_{i=2}^{k'} \text{Var}[Y_i - \mathbb{E}[Y_i|\mathbb{F}_i]|\mathbb{F}_i] \\
&\leq \frac{3000000\alpha}{C_1} \sum_{i=2}^{k'} (f_h^1(z_h^{i-1}) - f_h^2(z_h^{i-1}))^2 \\
&\leq \frac{3000000\alpha \cdot 2000000\alpha}{C_1} \\
&\leq \frac{(3000000\alpha)^2}{C_1}
\end{aligned}
$$

where the the first equality follows from the fact that for $i > k'$ the random variable is deterministic, the first inequality follows by the summation of the previous one and the second one by the fact that $\sum_{i=2}^{k'} (f_h^1(z_h^{i-1}) - f_h^2(z_h^{i-1}))^2 \leq \|f_h^1 - f_h^2\|_{\mathcal{Z}_h^{k'}} \leq Z_{k'}$.

We are now ready to use Freedman's inequality (Lemma C.6) with $R = \frac{3000000\alpha}{C_1}, \sigma^2 = \frac{(3000000\alpha)^2}{C_1}$. We get

$$\Pr\left(\left|\sum_{i=1}^{k}(Y_i - \mathbb{E}[Y_i|\mathbb{F}_i])\right| \geq \alpha/100\right) = \Pr\left(\left|\sum_{i=1}^{k'}(Y_i - \mathbb{E}[Y_i|\mathbb{F}_i])\right| \geq \alpha/100\right)$$

$$\leq 2\exp\left\{-\frac{(\alpha/100)^2/2}{(3000000\alpha)^2/C_1 + \alpha^2 3000000/300C_1}\right\}$$

$$= 2\exp\left\{-\frac{C_1}{20000(3000000 + 10000)}\right\}$$

$$= 2\exp\left\{-\frac{C\log(T\mathcal{N}(\mathcal{F}_h, \sqrt{\delta/64T^3})/\delta)}{20000(3000000 + 10000)}\right\}$$

$$= 2\exp\left\{-\frac{C(\log((T\mathcal{N}(\mathcal{F}_h, \sqrt{\delta/64T^3})/\delta)^2))}{40000(3000000 + 10000)}\right\}$$

$$\leq (\delta/64T^2)/(\mathcal{N}(\mathcal{F}_h, \sqrt{\delta/64T^3}))^2$$

for some choice of $C$. Now we can take a union bound over all the functions in the discretized set and conclude that with probability at least $1 - \delta/(64T^2)$ we have that

$$\left|\sum_{i=1}^{k}(Y_i - \mathbb{E}[Y_i|\mathbb{F}_i])\right| \leq \alpha/100$$

for all pairs of functions in this set. We condition on this event and on the event in Lemma C.7. We first show that when this event happens, we have that $\mathcal{C}_h^k(\alpha) \subseteq \widehat{\mathcal{C}}_h^k(\alpha)$. Consider $f_1, f_2 \in \mathcal{C}_h^k(\alpha)$. We know that there exist $f_1', f_2' \in \mathcal{C}(\mathcal{F}, \sqrt{\delta/(64T^3)}) \times \mathcal{C}(\mathcal{F}, \sqrt{\delta/(64T^3)})$ with $||f_1 - f_1'||_\infty, ||f_2 - f_2'||_\infty \leq \sqrt{\delta/64T^3}$. Hence, we get that

$$||f_1' - f_2'||_{\mathcal{Z}_h^k}^2 \leq \left(||f_1 - f_1'||_{\mathcal{Z}_h^k} + ||f_2 - f_2'||_{\mathcal{Z}_h^k} + ||f_1 - f_2||_{\mathcal{Z}_h^k}\right)^2$$

$$\leq \left(||f_1 - f_2||_{\mathcal{Z}_h^k} + 2\sqrt{\delta|\mathcal{Z}_h^k|/(64T^3)}\right)^2 \leq \alpha/50$$

We now consider the $Y_i$'s that are generated by $f_1', f_2'$. It holds that $||f_1' - f_2'||_{\mathcal{Z}_h^k}^2 \leq \alpha/50 \implies ||f_1' - f_2'||_{\widehat{\mathcal{Z}}_h^{k-1}}^2 \leq \alpha/50$. Since the event $\mathcal{E}_h^{k-1}$ happens it follows that $\min\{||f_1' - f_2'||_{\widehat{\mathcal{Z}}_h^{k-1}}^2, T(H+1)^2\} \leq 100(\alpha/50) = 2\alpha < 2000000\alpha \implies Z_k \leq 2000000\alpha$. Thus, every $Y_i$ is exactly $(f_1'(z_h^i) - f_2'(z_h^i))^2$ multiplied by the number of times $z_h^i$ is in the sub-sampled dataset. Hence, we get

$$||f_1' - f_2'||_{\widehat{\mathcal{Z}}_h^k}^2 = \sum_{i=2}^{k} Y_i \leq \sum_{i=2}^{k}\mathbb{E}[Y_i|\mathbb{F}_i] + \alpha/100$$

$$\leq ||f_1' - f_2'||_{\mathcal{Z}_h^k}^2 + \alpha/100 \leq 3\alpha/100$$

where the first inequality follows from the concentration bound we have derived and the other two simply from the definitions of these quantities.

We now bound $||f_1 - f_2||_{\widehat{\mathcal{Z}}_h^k}^2$. We have that

$$||f_1 - f_2||_{\widehat{\mathcal{Z}}_h^k}^2 \leq \left(||f_1' - f_2'||_{\widehat{\mathcal{Z}}_h^k} + ||f_1 - f_1'||_{\widehat{\mathcal{Z}}_h^k} + ||f_2 - f_2'||_{\widehat{\mathcal{Z}}_h^k}\right)^2$$

$$\leq (||f_1' - f_2'||_{\widehat{\mathcal{Z}}_h^k} + 2\sqrt{|\widehat{\mathcal{Z}}_h^k|} \cdot \sqrt{\delta/(64T^3)})^2$$

$$\leq (||f_1' - f_2'||_{\widehat{\mathcal{Z}}_h^k} + 2)^2 \leq (\sqrt{3\alpha/100} + 2)^2 \leq \alpha$$

Hence, we have shown that $\underline{\mathcal{C}}_h^k(\alpha) \subseteq \widehat{\mathcal{C}}_h^k(\alpha)$. So in this case, the one inequality that define $\mathcal{E}_h^k$ holds.

We shift our attention to the second inequality now. We will show the contrapositive of our claim, i.e. if $f_1, f_2 \notin \overline{C}_h^k(\alpha) \implies f_1, f_2 \notin \widehat{C}_h^k(\alpha)$. Let $f_1, f_2 \in \mathcal{F}_h \times \mathcal{F}_h$ such that $||f_1 - f_2||_{\mathcal{Z}_h^k} > 100\alpha$. We know that there exist $f_1', f_2' \in \mathcal{C}(\mathcal{F}, \sqrt{\delta/(64T^3)}) \times \mathcal{C}(\mathcal{F}, \sqrt{\delta/(64T^3)})$ with $||f_1 - f_1'||_\infty, ||f_2 - f_2'||_\infty \leq \sqrt{\delta/64T^3}$. Hence, using the triangle inequality we get that

$$||f_1' - f_2'||_{\mathcal{Z}_h^k}^2 \geq (||f_1 - f_2||_{\mathcal{Z}_h^k} - ||f_1 - f_1'||_{\mathcal{Z}_h^k} - ||f_2 - f_2'||_{\mathcal{Z}_h^k})^2$$
$$\geq (||f_1 - f_2||_{\mathcal{Z}_h^k} - 2\sqrt{|\mathcal{Z}_h^k|}\sqrt{\delta/(64T^3)})^2$$
$$= (\sqrt{100\alpha} - 2\sqrt{\delta/(64T^2)})^2 > 50\alpha$$

Again, consider the $Y_i$'s that are generated by $f_1', f_2'$. We want to show that $||f_1' - f_2'||_{\widehat{\mathcal{Z}}_h^k}^2 > 40\alpha$. Assume towards contradiction that $||f_1' - f_2'||_{\widehat{\mathcal{Z}}_h^k}^2 \leq 40\alpha$. We consider three different cases.

**First Case:** $||f_1' - f_2'||_{\mathcal{Z}_h^k}^2 \leq 2000000\alpha$. Similarly as before, we have that

$$||f_1' - f_2'||_{\widehat{\mathcal{Z}}_h^k}^2 = \sum_{i=2}^k Y_i \geq \mathbb{E}[Y_i|\mathbb{F}_i] - \alpha/100$$
$$> 50\alpha - \alpha/100 > 40\alpha$$

So we get a contradiction.

**Second Case:** $||f_1' - f_2'||_{\mathcal{Z}_h^{k-1}}^2 > 10000\alpha$. The contradiction comes directly from the fact that $\mathcal{E}_h^{k-1}$ holds, so
$$||f_1' - f_2'||_{\widehat{\mathcal{Z}}_h^k}^2 \geq ||f_1' - f_2'||_{\widehat{\mathcal{Z}}_h^{k-1}}^2 > 100\alpha$$
.

**Third Case:** $||f_1' - f_2'||_{\mathcal{Z}_h^{k-1}}^2 \leq 10000\alpha$ **and** $||f_1' - f_2'||_{\mathcal{Z}_h^k}^2 > 2000000\alpha$. We can directly see that for this case $(f_1'(z_h^k) - f_2'(z_h^k))^2 \geq 1900000\alpha$. Since $||f_1' - f_2'||_{\mathcal{Z}_h^{k-1}}^2 \leq 10000\alpha \implies ||f_1' - f_2'||_{\widehat{\mathcal{Z}}_h^{k-1}}^2 \leq 1000000\alpha$. Thus, since $\alpha \geq \beta$ we can see that the sensitivity is 1 so the element will be added to the sub-sampled dataset. Hence, $||f_1' - f_2'||_{\widehat{\mathcal{Z}}_h^k}^2 \geq (f_1'(z_h^k) - f_2'(z_h^k))^2 > 40\alpha$.

Thus, in any case we have that $||f_1' - f_2'||_{\widehat{\mathcal{Z}}_h^k}^2 > 40\alpha > \alpha$, so we get the result. $\qquad\square$

We are now ready to prove Lemma C.5.

*Proof of Lemma C.5:*

We know that for all $k \in [K], k \neq 1, h \in [H]$ it holds that

$$\Pr(\mathcal{E}_h^1 \mathcal{E}_h^2 \dots \mathcal{E}_h^{k-1}) - \Pr(\mathcal{E}_h^1 \mathcal{E}_h^2 \dots \mathcal{E}_h^k) = \Pr\left(\mathcal{E}_h^1 \mathcal{E}_h^2 \dots \mathcal{E}_h^{k-1}(\mathcal{E}_h^k)^c\right)$$
$$= \Pr\left(\mathcal{E}_h^1 \mathcal{E}_h^2 \dots \mathcal{E}_h^{k-1}\left(\cap_{n=0}^\infty \mathcal{E}_h^k(100^n \beta)\right)^c\right)$$
$$= \Pr\left(\mathcal{E}_h^1 \mathcal{E}_h^2 \dots \mathcal{E}_h^{k-1} \cup_{n=0}^\infty \mathcal{E}_h^k(100^n \beta)^c\right)$$
$$\leq \sum_{n=0}^\infty \Pr\left(\mathcal{E}_h^1 \mathcal{E}_h^2 \dots \mathcal{E}_h^{k-1}(\mathcal{E}_h^k(100^n \beta))^c\right)$$
$$= \sum_{n \geq 0, 100^n \beta \leq T(H+1)^2} \Pr\left(\mathcal{E}_h^1 \mathcal{E}_h^2 \dots \mathcal{E}_h^{k-1}(\mathcal{E}_h^k(100^n \beta))^c\right).$$

Thus, using Lemma C.9 we see that $\Pr(\mathcal{E}_h^1 \mathcal{E}_h^2 \dots \mathcal{E}_h^{k-1}) - \Pr(\mathcal{E}_h^1 \mathcal{E}_h^2 \dots \mathcal{E}_h^k) \leq \delta/(32T^2)(\log(T(H+1)^2/\beta) + 2) \leq \delta/32T$.

Hence, for any fixed $h \in [H]$ we get

$$\Pr\left(\bigcap_{k=1}^{K} \mathcal{E}_h^k\right) = 1 - \sum_{k=1}^{K} \left(\Pr(\mathcal{E}_h^1 \mathcal{E}_h^2 \ldots \mathcal{E}_h^{k-1}) - \Pr(\mathcal{E}_h^1 \mathcal{E}_h^2 \ldots \mathcal{E}_h^k)\right)$$
$$\geq 1 - K(\delta/32T) = 1 - \delta/(32H)$$

and by taking a union bound over $h \in [H]$ we get the result. $\square$

Now that we have shown that the sub-sampled dataset approximates well the original one, we shift our attention back to showing that our approach achieves optimism.

We first need a definition and a concetration lemma that is related to least-squares-estimators from prior work.

**Definition C.10.** A random variable $X$ is conditionally $\sigma$-subgaussian with respect to some filtration $\mathbb{F}$ if for all $\lambda \in \mathbb{R}$ it holds that $\mathbb{E}[\exp(\lambda X)] \leq \exp(\lambda^2 \sigma^2/2)$.

**Lemma C.11** ([RVR13], [AJS$^+$20]). *Let $\mathbb{F} = \{\mathbb{F}_p\}_{p=0,1,\ldots}$ be a filtration, $\{(X_p, Y_p)\}_p$ measurable random variables where $X_p \in \mathcal{X}, Y_p \in \mathbb{R}$. Let $\widetilde{\mathcal{F}}$ be a set of measurable functions from $\mathcal{X}$ to $\mathbb{R}$ and assume that $\mathbb{E}[Y_p | \mathbb{F}_{p-1}] = f^*(X_p)$ for some $f^* \in \widetilde{\mathcal{F}}$. Assume that $\{Y_p - f^*(X_p)\}_{p=1,\ldots}$ is conditionally $\sigma$-subgaussian given $\mathbb{F}_{p-1}$. Let $\widehat{f}_t = \arg\min_{f \in \widetilde{\mathcal{F}}} \sum_{p=1}^{t} (f(X_p) - Y_p)^2$ and $\widetilde{\mathcal{F}}_t(\beta) = \left\{f \in \widetilde{F} : \sum_{p=1}^{t} \left(f(X_p) - \widehat{f}(X_p)\right)^2 \leq \beta\right\}$. Then, for any $\alpha > 0$, with probability $1 - \delta$, for all $t \geq 1$ it holds that $f^* \in \widetilde{\mathcal{F}}_t(\beta_t(\delta, \alpha))$, where*

$$\beta_t(\delta, \alpha) = 8\sigma^2 \log(2\mathcal{N}(\widetilde{\mathcal{F}}, \alpha)/\delta) + 4t\alpha \left(C + \sqrt{\sigma^2 \log(4t(t+1)/\delta)}\right).$$

We are now ready to prove that our algorithm ensures optimism.

**Lemma C.12.** *With probability at least $1 - 2\delta$, we have that for all $h \in [H], k \in [K], s \in \mathcal{S}, a \in \mathcal{A}$*

$$Q_h^k(s,a) - Q_h^{\pi_k}(s,a) \leq \langle P_h(\cdot | s, a), V_{h+1}^k(\cdot) - V_{h+1}^{\pi_k}(\cdot)\rangle + 2b_h^k(s,a).$$

*Moreover, it holds that $Q_h^k(s,a) \geq Q_h^*(s,a)$.*

*Proof.* Fix some $h \in [H], k \in [K], s \in \mathcal{S}, a \in \mathcal{A}$. Throughout the proof, we condition on the events in Lemma C.5 and Lemma C.11. We assume that $k$ is a round that we perform an update.

We define $\mathcal{X} = \mathcal{S} \times \mathcal{A} \times \mathcal{V}, X_h^k = (s_h^k, a_h^k, V_{h+1}^k(\cdot)), Y_h^k = V_{h+1}^k(s_{h+1}^k)$. We also pick $\widetilde{\mathcal{F}} = \mathcal{F}_h$, where $\mathcal{F}_h$ is defined in Definition C.4. Then, we see that $\mathbb{E}[Y_h^k | \mathbb{F}_{k-1}] = f_h^*(X_h^k)$, where $f_h^*$ is the function that corresponds to the true model $P_h$, and we know that $f_h^* \in \mathcal{F}_h$. Recall that the optimization problem we solve in Algorithm 1 for every round $k$ we update our policy is

$$\widehat{P}_h^k = \arg\min_{P \in \mathcal{P}_h} \sum_{p=1}^{k} \left(\langle P(\cdot | s_h^p, a_h^p), V_{h+1}^p\rangle - V_{h+1}^p(s_{h+1}^p)\right)^2.$$

Based on the definition of $\mathcal{F}_h$, we can see that this is equivalent to

$$f_h^k = \arg\min_{f \in \mathcal{F}_h} \sum_{p=1}^{k} \left(f(s_h^p, a_h^p, V_{h+1}^p) - V_{h+1}^p(s_{h+1}^p)\right)^2.$$

Moreoever, $Y_h^k \in [0, H]$, so $Z_h^k = Y_h^k - f_h^*(X_h^k)$ is $H/2$-conditionally subgaussian. Thus, if we pick $\alpha = 1/T$ and

$$\beta_h^k = 4H^2 \log(2\mathcal{N}(\mathcal{F}_h, 1/T)/\delta) + 4k/T \left(C + \sqrt{H^2/4 \log(4(k(k+1)/\delta))}\right)$$

then Lemma C.11 gives us that $||f_h^* - \widehat{f}_h^k||_{\mathcal{Z}_h^k}^2 \leq \beta_h^k$. In particular, can pick

$$\widetilde{\beta} = \beta_h^K = 4H^2 \log(2\mathcal{N}(\mathcal{F}_h, 1/T)/\delta) + 4K/T \left(C + \sqrt{H^2/4 \log(4(K(K+1)/\delta))}\right)$$

and get a parameter that is independent of $k$. Moreover, Lemma C.5

$$||f_h^* - \widehat{f}_h^k||_{\widehat{\mathcal{Z}}_h^k} \le 100\widetilde{\beta} = \beta.$$

This implies that for our bonus function we have that $|\widehat{f}_h^k(s, a, V_{h+1}^k(\cdot)) - f_h^*(s, a, V_{h+1}^k(\cdot))| \le b_h^k(s, a)$.

Hence, we have that

$$\begin{aligned}
\langle \widehat{P}_h^k(\cdot|s, a), V_{h+1}^k(\cdot)\rangle - \langle P_h(\cdot|s, a), V_{h+1}^k(\cdot)\rangle &= \widehat{f}_h^k(s, a, V_{h+1}^k) - f_h^*(s, a, V_{h+1}^k) \\
&\le |\widehat{f}_h^k(s, a, V_{h+1}^k) - f_h^*(s, a, V_{h+1}^k)| \\
&\le b_h^k(s, a).
\end{aligned}$$

Now we use the definition of $Q_h^k(s, a), Q_h^{\pi_k}(s, a)$ to get that

$$\begin{aligned}
Q_h^k(s, a) &\le r_h^k(s, a) + \langle \widehat{P}_h^k(\cdot|s, a), V_{h+1}^k(\cdot)\rangle + b_h^k(s, a) \\
&\le r_h^k(s, a) + \langle P_h(\cdot|s, a), V_{h+1}^k(\cdot)\rangle + 2b_h^k(s, a) \\
Q_h^{\pi_k}(s, a) &= r_h^k(s, a) + \langle P_h(\cdot|s, a), V_{h+1}^{\pi_k}(\cdot)\rangle.
\end{aligned}$$

Combining these two, we get that

$$Q_h^k(s, a) - Q_h^{\pi_k}(s, a) \le \langle P_h(\cdot|s, a), V_{h+1}^k(\cdot) - V_{h+1}^{\pi_k}(\cdot)\rangle + 2b_h^k(s, a)$$

which proves the first part of the result.

For the second part, notice that if $Q_h^k(s, a) = H$ then the statement holds trivially since $Q_h^*(s, a) \le H$. So we can assume without loss of generality that $Q_h^k(s, a) = r_h^k(s, a) + \langle \widehat{P}_h^k(\cdot|s, a), V_{h+1}^k(\cdot)\rangle + b_h^k(s, a)$. The Bellman optimality condition gives us that $Q_h^*(s, a) = r_h^k(s, a) + \langle P_h^*(\cdot|s, a), V_{h+1}^*(\cdot)\rangle$. Hence, we have that

$$\begin{aligned}
Q_h^k(s, a) - Q_h^*(s, a) &= \langle \widehat{P}_h^k(\cdot|s, a), V_{h+1}^k(\cdot)\rangle - \langle P_h(\cdot|s, a), V_{h+1}^*(\cdot)\rangle + b_h^k(s, a) \\
&= \langle \widehat{P}_h^k(\cdot|s, a), V_{h+1}^k(\cdot)\rangle - \langle P_h(\cdot|s, a), V_{h+1}^k(\cdot)\rangle + \langle P_h(\cdot|s, a), V_{h+1}^k(\cdot)\rangle \\
&\quad - \langle P_h(\cdot|s, a), V_{h+1}^*(\cdot)\rangle + b_h^k(s, a) \\
&= \langle \widehat{P}_h^k(\cdot|s, a) - P_h(\cdot|s, a), V_{h+1}^k(\cdot)\rangle + \langle P_h(\cdot|s, a), V_{h+1}^k(\cdot) - V_{h+1}^*(\cdot)\rangle + b_h^k(s, a).
\end{aligned}$$

Now from our previous discussion it follows that $b_h^k(s, a) + \langle \widehat{P}_h^k(\cdot|s, a) - P_h(\cdot|s, a), V_{h+1}^k(\cdot)\rangle \ge 0$. Hence, it suffices to show that $\langle P_h(\cdot|s, a), V_{h+1}^k(\cdot) - V_{h+1}^*(\cdot)\rangle \ge 0$. To do that, we can just prove that $V_{h+1}^k(s') - V_{h+1}^*(s') \ge 0, \forall s' \in \mathcal{S}$. Since $V_{H+1}^k(s') = V_{H+1}^*(s') = 0, \forall s' \in \mathcal{S}$ we get that $Q_H^k(s, a) \ge Q_H^*(s, a)$. Thus, if we combine this with the update rule for $V_h^k, V_h^*$ we get the claim by induction. $\square$

Now that we have established the previous lemma, we need to bound the bonus that we are using in every round. The issue is that we do not update our policy in every round.

To do that, we follow a similar approach as in Appendix B.

**Lemma C.13.** *For every set $K' \subseteq [K]$ With probability at least $1 - \delta$, we have that*

$$\sum_{i=1}^{|K'|} \sum_{h=1}^{H} b_h^{k_i}(s_h^{k_i}, a_h^{k_i}) \le H + H(H+1)\dim_E(\mathcal{F}, 1/T) + CH\sqrt{\dim_E(\mathcal{F}, 1/T)|K'|\beta}$$

*where $\dim_E(\mathcal{F}, 1/T) = \max_{h \in [H]} \dim_E(\mathcal{F}_h, 1/T)$.*

*Proof.* We condition on the event described in Lemma C.5. From the definition of the bonus function, we have that for any $k \in [K]$

$$b_h^k(s_h^k, a_h^k) \le \bar{b}_h^k(s_h^k, a_h^k) = \sup_{||f_1 - f_2||^2_{\mathcal{Z}_h^k} \le 100\beta} |f_1(s_h^k, a_h^k) - f_2(s_h^k, a_h^k)|.$$

We bound $\sum_{i=1}^{K'} \bar{b}_h^{k_i}(s_h^{k_i}, a_h^{k_i})$ for each $h \in [H]$ separately.

Given some $\epsilon > 0$, we define $\widetilde{K} = \{k \in K' : \bar{b}_h^k(s_h^k, a_h^k) > \epsilon\}$, i.e. the set of episodes in $K'$ where the bonus function at $h$ has value greater than $\epsilon$. Consider some $k \in K'$. We denote $\mathcal{L}_h = \{(s_h^k, a_h^k, V_{h+1}^k(\cdot)) : k \in \widetilde{K}\}$, $L_h = |\mathcal{L}_h|$, and $N = L_h/\dim_E(\mathcal{F}_h, \epsilon) - 1$. Our goal is to show that there is some $z_h^k = (s_h^k, a_h^k, V_{h+1}^k(\cdot)) \in \mathcal{L}_h$ that is $\epsilon$-dependent on at least $N$ disjoint subsequences in $\mathscr{Z}_h^k \cap \mathcal{L}_h$.

To do that, we decompose $\mathcal{L}_h$ into $N + 1$ disjoint subsets and we denote the $j$-th subset by $\mathcal{L}_{h,j}$. We use the following procedure. Initially we set $\mathcal{L}_{h,j} = \emptyset$ for all $j \in [N + 1]$ and consider every $z_h^k \in \mathcal{L}_h$ in a sequential manner. For each such $z_h^k$ we find the smallest index $j, 1 \leq j \leq N$, such that $z_h^k$ is $\epsilon$-independent of the elements in $\mathcal{L}_{h,j}$ with respect to $\mathcal{F}_h$. If there is no such $j$, we set $j = N + 1$. Then, we update $\mathcal{L}_{h,j} \leftarrow \mathcal{L}_{h,j} \cup z_h^k$. Notice that after we go through all the elements of $\mathcal{L}_h$, we must have that $\mathcal{L}_{h,N+1} \neq \emptyset$. This is because every set $\mathcal{L}_{h,j}, 1 \leq j \leq N$, contains at most $\dim_E(\mathcal{F}_h, \epsilon)$ elements. Moreover, by definition, every element $z_h^k \in \mathcal{L}_{h,N+1}$ is $\epsilon$-dependent on at least $N$ disjoint subsequences in $\mathcal{L}_h$.

Furthermore, since $\bar{b}_h^k(s_h^k, a_h^k) > \epsilon$ for all $z_h^k \in \mathcal{L}_h$ there must exist $f_1, f_2 \in \mathcal{F}_h$ such that $|f_1(s_h^k, a_h^k, V_{h+1}^k(\cdot)) - f_2(s_h^k, a_h^k, V_{h+1}^k(\cdot))| > \epsilon$ and $||f_1 - f_2||_{\mathscr{Z}_h^k}^2 \leq 100\beta$. Hence, since $z_h^k \in \mathcal{L}_{h,N+1}$ is $\epsilon$-dependent on $N$ disjoint subsequences $\mathcal{L}_h$ and for each such subsequence $\mathcal{L}$, by the definition of $\epsilon$-dependence, it holds that $||f_1 - f_2||_{\mathcal{L}}^2 > \epsilon^2$ we have that

$$N\epsilon^2 \leq ||f_1 - f_2||_{\mathscr{Z}_h^k}^2 \leq 100\beta$$
$$\implies (L_h/\dim_E(\mathcal{F}_h, \epsilon) - 1)\epsilon^2 \leq 100\beta$$
$$\implies L_h \leq \left(\frac{100\beta}{\epsilon^2} + 1\right)\dim_E(\mathcal{F}_h, \epsilon).$$

We now pick a permutation $\bar{b}_1 \geq \bar{b}_2 \geq \ldots \geq \bar{b}_{|K'|}$ of the bonus functions $\{\bar{b}_h^k(s_h^k, a_h^k)\}_{k \in K'}$. For all $\bar{b}_k \geq 1/|K'|$ it holds that

$$k \leq \left(\frac{100\beta}{\bar{b}_k^2} + 1\right)\dim_E(\mathcal{F}_h, \bar{b}_k) \leq \left(\frac{100\beta}{\bar{b}_k^2} + 1\right)\dim_E(\mathcal{F}_h, 1/K') \implies$$
$$\bar{b}_k \leq \left(\frac{k}{\dim_E(\mathcal{F}_h, 1/K')} - 1\right)^{-1/2}\sqrt{100\beta}.$$

Moreover, notice that we get by definition that $\bar{b}_k \leq H + 1$. Hence, we have that

$$\sum_{i=1}^{|K'|} \bar{b}_h^{k_i}(s_h^{k_i}, a_h^{k_i}) = \sum_{i:\bar{b}_{k_i} < 1/|K'|} \bar{b}_h^{k_i}(s_h^{k_i}, a_h^{k_i}) + \sum_{i:\bar{b}_{k_i} \geq 1/|K'|} \bar{b}_h^{k_i}(s_h^{k_i}, a_h^{k_i})$$
$$\leq |K'| \cdot 1/|K'| + \sum_{i:\bar{b}_{k_i} \geq 1/|K'|, i \leq \dim_E(\mathcal{F}_h, 1/|K'|)} \bar{b}_h^{k_i}(s_h^{k_i}, a_h^{k_i})$$
$$+ \sum_{i:\bar{b}_{k_i} \geq 1/|K'|, |K'| \geq i > \dim_E(\mathcal{F}_h, 1/|K'|)} \bar{b}_h^{k_i}(s_h^{k_i}, a_h^{k_i})$$
$$\leq 1 + (H + 1)\dim_E(\mathcal{F}_h, 1/|K'|) + \sum_{|K'| \geq i > \dim_E(\mathcal{F}_h, 1/|K'|)} \left(\frac{k}{\dim_E(\mathcal{F}_h, 1/|K'|)} - 1\right)^{-1/2}\sqrt{100\beta}$$
$$\leq 1 + (H + 1)\dim_E(\mathcal{F}_h, 1/|K'|) + C\sqrt{\dim_E(\mathcal{F}_h, 1/|K'|)|K'|\beta}$$
$$\leq 1 + (H + 1)\dim_E(\mathcal{F}_h, 1/T) + C\sqrt{\dim_E(\mathcal{F}_h, 1/T)|K'|\beta}$$

for some constant $C > 0$, where the second to last inequality can be obtained by bounding the summation by the integral and the last one by the definition of the eluder dimension. We get the final result by summing up all the inequalities over $H$. $\qquad\square$

The next step in our proof, is to bound the number of episodes that our policy can be worse than the optimal one by $2^n \text{gap}_{\min}$, for all $n \in \mathbb{N}$. This is inspired by [HZG21].

**Lemma C.14.** *If we pick*

$$\beta = 4H^2 \log(2\mathcal{N}(\mathcal{F}, 1/T)/\delta) + 4/H\left(C + \sqrt{H^2/4 \log(4(K(K+1)/\delta))}\right),$$

*then for every $h \in [H]$ and $n \in \mathbb{N}$, with probability at least $1 - (K+3)\delta$, we have that*

$$\sum_{k=1}^{K} \mathbb{1}\left[V_h^*(s_h^k) - Q_h^{\pi_k}(s_h^k, a_h^k) \geq 2^n \text{gap}_{\min}\right] \leq \frac{\widetilde{C}H^4 \log(T\mathcal{N}(\mathcal{F}_h, 1/T)/\delta) \dim_E^2(\mathcal{F}, 1/T)}{4^n \text{gap}_{\min}^2}.$$

*Proof.* We keep $h$ fixed.

We denote by $K'$ the set of episodes where the gap at step $h$ is at least $2^n$, i.e.

$$K' = \left\{k \in [K] : V_h^*(s_h^k) - Q_h^{\pi_k}(s_h^k, a_h^k) \geq 2^n \text{gap}_{\min}\right\}.$$

The goal is to bound the quantity $\sum_{i=1}^{|K'|}\left(Q_h^{k_i}(s_h^{k_i}, a_h^{k_i}) - Q_h^{\pi_{k_i}}(s_h^{k_i}, a_h^{k_i})\right)$ from below and above with functions $f_1(|K'|), f_2(|K'|)$ and then use the fact that $f_1(|K'|) \leq f_2(|K'|)$ to derive an upper bound on $|K'|$.

For the lower bound, we have that

$$\sum_{i=1}^{|K'|}\left(Q_h^{k_i}(s_h^{k_i}, a_h^{k_i}) - Q_h^{\pi_{k_i}}(s_h^{k_i}, a_h^{k_i})\right) \geq \sum_{i=1}^{|K'|}\left(Q_h^{k_i}(s_h^{k_i}, \pi_h^*(s_h^{k_i})) - Q_h^{\pi_{k_i}}(s_h^{k_i}, a_h^{k_i})\right)$$

$$\geq \sum_{i=1}^{|K'|}\left(Q_h^*(s_h^{k_i}, \pi_h^*(s_h^{k_i})) - Q_h^{\pi_{k_i}}(s_h^{k_i}, a_h^{k_i})\right)$$

$$= \sum_{i=1}^{|K'|}\left(V_h^*(s_h^{k_i}) - Q_h^{\pi_{k_i}}(s_h^{k_i}, a_h^{k_i})\right)$$

$$\geq 2^n \text{gap}_{\min}|K'|$$

where the first inequality holds by the definition of the policy $\pi_{k_i}$, the second one follows because $Q_h^{k_i}(\cdot, \cdot)$ is an optimistic estimate of $Q_h^*(\cdot, \cdot)$ which happens with probability at least $1 - 2\delta$ (see Lemma C.12) and the third one by the definition of $k_i$.

We get the upper bound on this quantity in the following way. For any $h' \in [H]$ we have

$$Q_{h'}^k(s_{h'}^k, a_{h'}^k) - Q_{h'}^{\pi_k}(s_{h'}^k, a_{h'}^k) \leq \sum_{s' \in \mathcal{S}} P_{h'}(s'|s_{h'}^k, a_{h'}^k)V_{h'+1}^k(s') + r_{h'}(s_{h'}^k, a_{h'}^k) + 2b_{h'}^k(s_{h'}^k, a_{h'}^k) - Q_{h'}^{\pi_k}(s_{h'}^k, a_{h'}^k)$$

$$= \left\langle P_{h'}(\cdot|s_{h'}^k, a_{h'}^k), V_{h'+1}^k - V_{h'+1}^{\pi_k}\right\rangle + 2b_{h'}^k(s_{h'}^k, a_{h'}^k)$$

$$= V_{h'+1}^k(s_{h'+1}^k) - V_{h'+1}^{\pi_k}(s_{h'+1}^k) + \epsilon_{h'}^k + 2b_{h'}^k(s_{h'}^k, a_{h'}^k)$$

$$= Q_{h'+1}^k(s_{h'+1}^k, a_{h'+1}^k) - Q_{h'+1}^{\pi_k}(s_{h'+1}^k, a_{h'+1}^k) + \epsilon_{h'}^k + 2b_{h'}^k(s_{h'}^k, a_{h'}^k)$$

where we define $\epsilon_{h'}^k = \left\langle P_{h'}(\cdot|s_{h'}^k, a_{h'}^k), V_{h'+1}^k - V_{h'+1}^{\pi_k}\right\rangle - \left(V_{h'+1}^k(s_{h'+1}^k) - V_{h'+1}^{\pi_k}(s_{h'+1}^k)\right)$ and the inequality follows from Lemma C.12.

We now take the summation over all $k \in |K'|, h \leq h' \leq H$ and we get

$$\sum_{i=1}^{|K'|}\sum_{h'=h}^{H}\left(Q_h^{k_i}(s_h^{k_i}, a_h^{k_i}) - Q_h^{\pi_{k_i}}(s_h^{k_i}, a_h^{k_i})\right) \leq \sum_{i=1}^{|K'|}\sum_{h'=h}^{H}\epsilon_{h'}^{k_i} + \sum_{i=1}^{|K'|}\sum_{h'=h}^{H}b_{h'}^{k_i}(s_{h'}^{k_i}, a_{h'}^{k_i}).$$

We will bound each of the two terms on the RHS separately.

For the first term, we notice that $x_j = \left\langle P_j(\cdot|s_j^{k_i}, a_j^{k_i}), V_{j+1}^{k_i} - V_{j+1}^{\pi_{k_i}}\right\rangle - \left(V_{j+1}^{k_i}(s_{j+1}^{k_i}) - V_{j+1}^{\pi_{k_i}}(s_{j+1}^{k_i})\right)$ forms a martingale difference sequence with zero mean and

$|x_j| \le 2H$. Hence, we can use Lemma B.7 and that for each $k \in K'$, with probability at least $1 - \delta$ we have that

$$\sum_{i=1}^{k} \sum_{j=1}^{H} \left( \left\langle P_j(\cdot | s_j^{k_i}, a_j^{k_i}), V_{j+1}^{k_i} - V_{j+1}^{\pi_{k_i}} \right\rangle - \left( V_{j+1}^{k_i}(s_{j+1}^{k_i}) - V_{j+1}^{\pi_{k_i}}(s_{j+1}^{k_i}) \right) \right) \le \sqrt{8kH^3 \log(1/\delta)}.$$

If we take the union bound over all $k \in [K]$ we have that with probability at least $1 - |K'|\delta$

$$\sum_{i=1}^{|K'|} \sum_{h'=h}^{H} \epsilon_{h'}^{k_i} \le \sqrt{8|K'|H^3 \log(1/\delta)}.$$

We now focus on the second term. Using Lemma C.13 we get that

$$\sum_{i=1}^{|K'|} \sum_{h'=h}^{H} b_{h'}^{k_i}(s_{h'}^{k_i}, a_{h'}^{k_i}) \le H + H(H+1) \dim_E(\mathcal{F}, 1/T) + CH\sqrt{\dim_E(\mathcal{F}, 1/T)|K'|\beta}$$

and this happens with probability at least $1 - \delta$. Hence, combining the upper and lower bound of

$$\sum_{i=1}^{|K'|} \left( Q_h^{k_i}(s_h^{k_i}, a_h^{k_i}) - Q_h^{\pi_{k_i}}(s_h^{k_i}, a_h^{k_i}) \right)$$

we get that

$$2^n \mathrm{gap}_{\min} |K'| \le \sqrt{8|K'|H^3 \log(1/\delta)} + H + H(H+1)\dim_E(\mathcal{F}, 1/T) + CH\sqrt{\dim_E(\mathcal{F}, 1/T)|K'|\beta}$$

$$\implies |K'| \le \frac{\widetilde{C}H^4 \log(T\mathcal{N}(\mathcal{F}, 1/T)/\delta) \dim_E^2(\mathcal{F}, 1/T)}{4^n \mathrm{gap}_{\min}^2}.$$

$\square$

We are now ready to prove Lemma C.1.

*Proof of Lemma C.1:* Throughout this proof we condition on the event described in Lemma C.14 which happens with probability at least $1 - (K+3)\delta$. Since $\mathrm{gap}_{\min} > 0$ whenever we do not take the optimal action, we have that either $V_h^*(s_k) - Q_h^*(s_h^k, a_h^k) = 0$ or $V_h^*(s_k) - Q_h^*(s_h^k, a_h^k) \ge \mathrm{gap}_{\min}$. Our approach is to divide the interval $[0, H]$ into $N = \lceil \log(H/\mathrm{gap}_{\min}) \rceil$ intervals and count the number of $V_h^*(s_k) - Q_h^*(s_h^k, a_h^k)$ that fall into each interval. Notice that for every $V_h^*(s_k) - Q_h^*(s_h^k, a_h^k)$ that falls into interval $i$ we can get an upper bound of $V_h^*(s_k) - Q_h^*(s_h^k, a_h^k) \le 2^i \mathrm{gap}_{\min}$ and this upper bound is essentially tight. Hence, we have that

$$\begin{aligned}
\sum_{k=1}^{K} \left( V_h^*(s_h^k) - Q_h^*(s_h^k, a_h^k) \right) &\le \sum_{i=1}^{N} \sum_{k=1}^{K} \mathbb{1}\left[ 2^i \mathrm{gap}_{\min} \ge V_h^*(s_h^k) - Q_h^*(s_h^k, a_h^k) \ge 2^{i-1}\mathrm{gap}_{\min} \right] \cdot 2^i \mathrm{gap}_{\min} \\
&\le \sum_{i=1}^{N} \sum_{k=1}^{K} \mathbb{1}\left[ V_h^*(s_h^k) - Q_h^*(s_h^k, a_h^k) \ge 2^{i-1}\mathrm{gap}_{\min} \right] \cdot 2^i \mathrm{gap}_{\min} \\
&\le \sum_{i=1}^{N} \frac{\widetilde{C}H^6 \log(T\mathcal{N}(\mathcal{F}, 1/T)/\delta) \dim_E^2(\mathcal{F}, 1/T)}{4^{i-1}\mathrm{gap}_{\min}^2} \cdot 2^i \mathrm{gap}_{\min} \\
&= \sum_{i=1}^{N} \frac{C'H^4 \log(T\mathcal{N}(\mathcal{F}, 1/T)/\delta) \dim_E^2(\mathcal{F}, 1/T)}{2^i \mathrm{gap}_{\min}} \\
&\le \frac{CH^4 \log(T\mathcal{N}(\mathcal{F}, 1/T)/\delta) \dim_E^2(\mathcal{F}, 1/T)}{\mathrm{gap}_{\min}}
\end{aligned}$$

where the first inequality holds by the definition of the intervals, the second due to the properties of the indicator function, the third because of Lemma C.14 and in the last two steps we just manipulate the constants. $\square$