# OpenReview forum: "Reinforcement Learning with Logarithmic Regret and Policy Switches"
_NeurIPS.cc/2022/Conference — NeurIPS 2022 Accept_

### Official Review · Reviewer_Bt98 · 2022-07-10

**Rating:** 8
**Confidence:** 5
**Soundness:** 4 excellent
**Presentation:** 4 excellent
**Contribution:** 4 excellent

**Summary:**

Regret in the RL setting has been extensively studied for linear approximations but not for other families of functions. In addition, it is more typical to have episode specific policies which is clearly undesirable. The authors study regret for a general set of functions and with a switching cost of policies among episodes. They exhibit an algorithm with optimal regret by showing an upper bound on regret of their algorithm and a lower bound of the same order.

Minor typos: line 57: to model-based
line 129: of the optimal policy

**Questions:**

Theorems 3.1 and 3.2 use tau and delta. The statements hold for any non-negative values of these two parameters. It's never discussed what value should they have in order to minimize the regret bound.

**Limitations:**

This being analytical work, there are no clear societal negative impacts.

**Strengths And Weaknesses:**

Although this is an analytical paper, it would be great to showcase practical performance of their algorithm.
The introduction is slightly misleading as it doesn't mention that the regret bounds are in the probabilistic sense.

The paper has many contributions, including the algorithm and the underlying non-trivial analyses.

---

> ### Author Response · Authors · 2022-08-02
> **Values of $\tau, \delta$**
>
> We would like to thank the reviewer for appreciating our paper and for mentioning that our work has many contributions.
>
> We will fix the typos that were pointed out and mention that the regret bound is probabilistic.
>
> As you correctly point out, the main focus of our work is the theoretical analysis of the algorithms we present and the analytical properties they satisfy. Nevertheless, we will try to include the empirical performance of our algorithm in the next version of our manuscript.
>
> Thank you for pointing out that we forgot to mention what values of $\delta, \tau$ satisfy the logarithmic regret guarantees we are aiming for. A choice that works is setting $\delta = 1/T, \tau = \log T$. We will mention it in the next version of our manuscript.

---

### Official Review · Reviewer_HxV2 · 2022-07-11

**Rating:** 4
**Confidence:** 3
**Soundness:** 3 good
**Presentation:** 3 good
**Contribution:** 2 fair

**Summary:**

The paper studies the problem of regret minimization for episodic RL with general function approximation. Under both model-free and model-based settings, logarithmic regret and policy switch results are presented.

**Questions:**

Could you elaborate on the technical improvements made in this paper over previous results? I would also suggest discuss Theorem 3.1 and Theorem 3.2 in context of previous results of log regret for RL under tabular/linear function approximations.

**Limitations:**

The authors discussed this and I agree that this is a theoretical work, which has no potential societal impact.

**Strengths And Weaknesses:**

Pros: The paper is very well written, and all the results are presented clearly. The background and the related works of this work are also discussed sufficiently. The results presented are new and important to RL as well.

Cons: The results are not very surprising given the prior works on RL with logarithmic regret and RL with general function approximation. The peeling techniques for regret decomposition are also used to obtain logarithmic regret for RL under linear function approximation. The other parts of the proof for the main theorem and the sub-sampling method mainly follow from results for RL under general function approximation [KSWY21]. By the property of the sub-sampled dataset, and as Lemma C.3 pointed out, logarithmic policy switching is a direct result.

---

> ### Author Response · Authors · 2022-08-02
> **Technical Improvements, Comparison with Prior Results**
>
> We would like to thank the reviewer for finding our paper very well-written and for mentioning that our results are important to the RL community.
>
> (**Technical improvements**) We want to now comment on the improvements of our work compared to the literature as well as the technical challenges in establishing our results.
>
> - *Model-based setting*: Firstly, we underline that, even though our algorithm in the model-based setting has a similar flavor to the one in [KSWY21], the analysis of the algorithm is different. Εven the way we define the bonus makes the setting more complicated since we have to keep track of $(s,a,V)$ tuples and not only state-action pairs as in the model-free setting. The bulk of our technical contribution is the proof of Lemma C.1. As you correctly pointed out, if one has shown Lemma C.1, then the bound on the adaptivity complexity follows from the sub-sampling procedure. However, prior to our work it was not clear that something similar to Lemma C.1 can be shown using sub-sampling in the model-based setting.
> To establish this lemma we have proven a series of technical lemmas: Lemma C.9, Lemma C.5, Lemma C.12, Lemma C.13, Lemma C.14. Essentially, in Lemma C.5 and Lemma C.9 we show that the model-based sampling procedure that we have defined creates a dataset which "approximates" the original one. Subsequently, Lemma C.12 shows optimism using a different approach than the one in [KSWY21]. Importantly, since neither the model-free nor the model-based assumption implies the other, one cannot use the approach in [KSWY21] to prove optimism. Next, in Lemma C.13 we show how to handle the summation of the bonus we add. Finally, Lemma C.14 is the only technical lemma in which the minimum sub-optimality gap assumption and the analysis using the peeling technique are needed.
>
> - *Model-based algorithm, instance-independent regret*: As we mentioned in the paper (we kindly refer to lines 314-316), if we drop the minimum-suboptimality gap assumption and use the standard regret decomposition, e.g. the one that appears in [AJS+20], we can get an instance-independent $\sqrt{T}$-regret bound in the model-based setting using only $\text{poly}\log T$ policy updates. Prior to our work, it was unclear from the works of [KSWY21], [AJS+20] that the sub-sampling approach could work in the model-based setting as well. Importantly, this provides an exponential improvement over the algorithm in [AJS+20].
>
> - *Model-free setting*: In the model-free setting, the key technical lemma that we need is Lemma B.2. To establish this, two key lemmas are Lemma B.6 and Lemma B.9. Essentially, Lemma B.6 generalizes a result of [KSWY21] which bounds the summation of the bonus functions over all the episodes $K$, to a bound of the summation of the bonus functions over any set of rounds $K’ \subseteq [K]$. In Lemma B.9, we utilize the peeling technique to establish a bound similar to the one that appears in [HZG21], which only holds for linear functions.
>
> - *Unified approach*: We view the fact that our treatment of both settings is of a similar flavor and, essentially, unified, as an advantage of our work which can shed more light into the connection and the interplay between the two settings and inspire future research direction.
>
>
> (**Comparison with prior results**) We remark that we can recover the results of [HZG’21], which focuses on the linear setting, up to $\text{poly}\log T$ factors. Moreover, the algorithms we use achieve an exponential decrease in the adaptivity complexity compared to the algorithms that are used in this work. Furthermore, it is not clear how to adapt techniques that are used to show optimism and to bound the summation of the bonus functions from the linear setting to the general function approximation setting. Thus, moving to general function approximation is non-trivial. We also remark that, to the best of our knowledge, even if one focuses just on the linear setting, our work is the first one which establishes logarithmic regret guarantees and logarithmic adaptivity in the model-free and the model-based settings.

---

### Official Review · Reviewer_jPGw · 2022-07-11

**Rating:** 6
**Confidence:** 3
**Soundness:** 3 good
**Presentation:** 4 excellent
**Contribution:** 3 good

**Summary:**

The paper considers the reinforcement learning problem with
general function classes and general model classes. In a specific
setting with a strictly positive optimality gap, the authors show
logarithmic regret can be achieved (in contrast to the minimax
optimal regret $\sqrt{T}$). Moreover, the proposed algorithm
only switches policies for polylog times. The results are established
in both the model-based and model-free settings.

**Questions:**

In bandit problems, the optimal number of batches/switches
is of the order $\log\log{T}$. Is the number of switches $\log{T}$ optimal
in RL, or is it the same as in the bandit problem?


**Limitations:**

The authors have adequately addressed the limitations and potential negative societal impact of their work.

**Strengths And Weaknesses:**

1. The paper is well-written, with an extensive literature review
and a clear exposition of the proposed method; the analysis of
the algorithm is complemented with explanation and intuitions.

2. Technically, the paper is solid to my knowledge and the extension of the method is nontrivial.

3. It might help if this paper could show numerical results
illustrating the claims made in theories.

---

> ### Author Response · Authors · 2022-08-02
> **Experiments, Optimal Number of Switches**
>
> We would like to thank the reviewer for finding our paper well-written and for mentioning that the extensions we propose are non-trivial.
>
> (**Experiments**) We want to underline that the main focus of our work is the theoretical analysis of the algorithms we present and the analytical properties they satisfy. Nevertheless, if the reviewer finds it useful, we can include the empirical performance of our algorithm in the next version of our manuscript.
>
> (**Optimal number of switches**) The question regarding the optimal number of switches that you raised is a very interesting one and an important future research direction. We remark that the $\log\log T$ bound is the optimal number of switches in bandits when the contexts are *stochastic*. When the contexts are adversarial, the lower bound is $\Omega(\log T)$*. We conjecture that $O(\log T)$ is the correct bound since RL is more akin to adversarial contexts. We also underline that even in the linear function approximation setting and the tabular setting in RL there are no known algorithms that improve upon $O(\log T)$ switches. However, all the RL papers with low adaptivity complexity that we are aware of do not provide any lower bounds in terms of the number of episodes $K$, even in the linear function approximation setting or the tabular setting. It is known that the lower bound has to be polynomial in $H$ but a characterization on the dependence on $K$ remains elusive. Moreover, it is not clear to us how to adapt the approach in the bandit setting to, potentially, show a lower bound of $\Omega(\log T)$ in the RL setting. We hope and believe that the line of work that our paper is a part of will inspire future research directions like the one you mention.
>
> *Linear Bandits with Limited Adaptivity and Learning Distributional Optimal Design, Yufei Ruan, Jiaqi Yang, Yuan Zhou

---

> > ### Comment · Reviewer_jPGw · 2022-08-08
> > **Response**
> >
> > I thank the authors for addressing my questions. I remain positive about this paper.

---

### Official Review · Reviewer_PwVr · 2022-07-11

**Rating:** 7
**Confidence:** 3
**Soundness:** 3 good
**Presentation:** 4 excellent
**Contribution:** 3 good

**Summary:**

The paper studies the instance-dependent logarithmic regret bounds on the RL problem with general function approximation. The paper introduces the notation of “adaptivity complexity”. The paper designs two RL algorithms with general function approximation separately in model-based and model-free settings that archive the logarithmic regret bounds and logarithmic adaptivity complexity simultaneously.

**Questions:**

I think this paper is somewhat missing an assessment of the tractability of the proposed algorithm, especially the computational complexity of the optimization problems involved in Algorithms 2 and 3. Can the author clarify this?

**Limitations:**

The assumptions and limitations are stated in Section 2.

**Strengths And Weaknesses:**

Logarithmic instance-dependent regret bounds and logarithmic switching cost has been studied in tabular RL and RL with linear function approximation. The idea is not novel, but this paper is the first work that establishes a logarithmic instance-dependent regret and logarithmic switching cost guarantee for RL with general function approximation. This is a good paper. The paper is well-written and clearly delivers the message. Related work is covered well. The paper makes strong use of the theoretical results published in a previous paper [KSWY21] but presents novel and important generalizations to model-based RL.

---

> ### Author Response · Authors · 2022-08-02
> **Computational Complexity**
>
> We would like to thank the reviewer for finding our paper well-written and for finding our model-based algorithm novel and important. In the following, we comment on the computational tractability of the algorithm we propose.
>
> As it argued in  [KSWY21], the main computational bottleneck of the algorithms is solving a (weighted) least squares problem over some set. Given such a procedure, one can also construct the confidence region and estimate the exploration bonus and the sensitivity. [KSWY21] shows that the computational complexity of the algorithm overall is $O(\text{poly}(d_{\mathcal{F}}H \log K))$. Importantly, if there is some structure on the space and solving the least-squares problem becomes easier then the computational complexity of the algorithm drops. We underline that solving a least-squares problem is an important component of most theoretical RL algorithms that we are aware of which go beyond the tabular setting. A similar result holds in the model-based setting as well. We will include a discussion about the running time in the next version of our manuscript.

---

### Meta-Review · Area_Chair_fDuV · 2022-08-25

**Recommendation:** Accept
**Confidence:** Certain

**Metareview:**

Reviewers are generally positive about the paper and I see that this paper's techniques are differentiated from KSWY 21. Please make sure you address all the reviewers' comments and incorporate them (and any new experimental results, if applicable) in your camera-ready.

**Award:**

No

---

### Decision · Program_Chairs · 2022-09-14

Accept